# LANGUAGE-INFORMED VISUAL CONCEPT LEARNING

**Sharon Lee**[*]   **Yunzhi Zhang**[*]   **Shangzhe Wu**   **Jiajun Wu**
Stanford University

## ABSTRACT

Our understanding of the visual world is centered around various concept axes, characterizing different aspects of visual entities. While different concept axes can be easily specified by language, *e.g.*, `color`, the exact visual nuances along each axis often exceed the limitations of linguistic articulations, *e.g.*, a particular style of painting. In this work, our goal is to learn a language-informed visual concept representation, by simply distilling large pre-trained vision-language models. Specifically, we train a set of concept encoders to encode the information pertinent to a set of language-informed concept axes, with an objective of reproducing the input image through a pre-trained Text-to-Image (T2I) model. To encourage better disentanglement of different concept encoders, we anchor the concept embeddings to a set of text embeddings obtained from a pre-trained Visual Question Answering (VQA) model. At inference time, the model extracts concept embeddings along various axes from new test images, which can be remixed to generate images with novel compositions of visual concepts. With a lightweight test-time finetuning procedure, it can also generalize to novel concepts *unseen* at training. Project page at `https://cs.stanford.edu/~yzzhang/projects/concept-axes`.

## 1 INTRODUCTION

In order to make sense of the myriad visual entities in the world, humans develop an abstracted generative model of them and organize the underlying sources of variation into *visual concepts*, such as different colors or different types of objects. Designing systems that can recognize visual concepts within images as humans do has been a longstanding goal in the fields of computer vision and artificial intelligence (Russakovsky et al., 2015; Krizhevsky et al., 2012; Girshick et al., 2014).

To facilitate efficient reasoning and communication of these concepts, humans created symbolic depictions that have evolved into natural language. Such natural language grounding of visual data has been instrumental in the recent proliferation of powerful large vision-language models that are capable of semantically identifying objects in images (Radford et al., 2021; Kirillov et al., 2023) or generating photo-realistic images from arbitrary text prompts (Ramesh et al., 2021; Rombach et al., 2022; Saharia et al., 2022; Yu et al., 2022). While different *concept axes* can be easily specified by words, such as `category` and `style`, it is much less intuitive to delineate the subtleties of low-level visual nuances along each axis using language, such as *one particular style* of a painting.

In this work, our goal is to distill from large pre-trained vision-language models a function that extracts visual concepts along a set of language-specified concept axes from images. As illustrated in Figure 1, once these concepts are extracted, we can recompose them across different image instances at inference time to produce new images with novel concept combinations. To learn this function, rather than collecting a large-scale dataset of human annotations for each specific visual concept, we design a language-informed visual concept representation, and simply distill from a pre-trained Text-to-Image (T2I) generation model. There are three fundamental properties we seek in this visual concept representation.

First, unlike T2I generation, which relies on generic words as visual concept descriptors, we would like to capture fine-grained visual nuances using continuous concept embeddings. One common technique is to invert the text-to-image generation process by optimizing an embedding with the objective of reproducing a given input image using a pre-trained T2I model, often referred to as Textual Inversion (Gal et al., 2022). However, most existing Textual Inversion methods (Gal et al.,

---

[*]Equal contribution; alphabetically ordered.

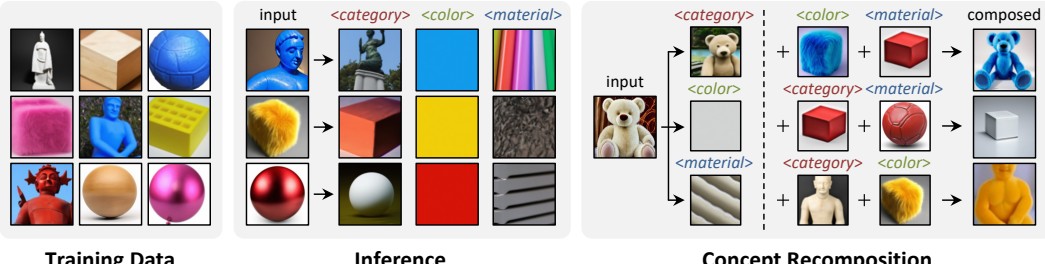

Figure 1: **Language-Informed Visual Concept Learning.** Our goal is to learn a visual concept representation grounded on a set of language-informed concept axes, *e.g.*,category, color, and material, by simply distilling from pre-trained text-to-image generation models without manual annotations. After training, the concept encoders extract *disentangled* axis-specific embeddings from an image, which can be remixed to generate new images with novel concept compositions.

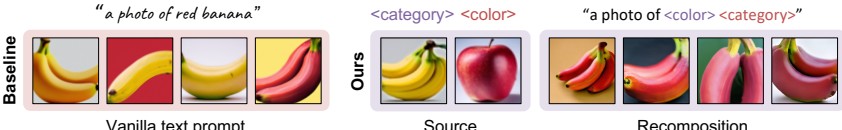

Figure 2: **Learned Disentangled Concept Embeddings Improve Compositionality.** Left: Vanilla text-to-image model may fail to adhere to text prompts of uncommon combinations of concepts even with prompt engineering, *e.g.* "red banana". Right: With the same backbone T2I generator, our learned disentangled concept embeddings greatly enhance concept compositionality.

2022) optimize embeddings for individual image instances independently, overlooking the shared nature of visual concepts across instances. For instance, the concept of "red" is shared between a "red apple" and a "red dress". Moreover, the concepts of "red" and "yellow" also are instances of the property of color.

Hence, the second desired property of the visual concept representation is to preserve such common concept structures among various visual instances. Instead of optimizing on individual image instances independently, we design a set of *concept encoders*, where each encoder learns to encode the visual characteristics of an input image pertaining to one concept axis specified by language. This ensures that the inverted concept embeddings can be shared across different instances and remixed to generate new images.

The third crucial aspect of this representation is to ascertain that different concept axes are disentangled, allowing for changes to be made specifically on single concept axis without modifying other axes. To do so, we reuse the disentangled nature of linguistic concepts and ground the predictions to a set of discrete text anchors in the concept embeddings space, which can be obtained by querying a pre-trained generic Visual Question Answering (VQA) model, *e.g.*, BLIP-2 (Li et al., 2023b). This soft anchoring constraint significantly improves the disentanglement of concept embeddings across different axes while still retaining sufficient leeway to capture nuanced visual variations that BLIP-2 struggles to discern, *e.g.*, the style of an art piece in Figure 6.

Putting these ideas together, we design a generic framework for learning disentangled and compositional visual concepts grounded to linguistic structures by exploiting pre-trained text-to-image generation and visual question answering models. We show that these concept encoders can be trained purely on synthetic images generated by a pre-trained T2I model, and extract concept embeddings from real images at test time, which capture the fine-grained visual nuances.

Our contributions can be summarized as follows:

1. We propose a generic framework for learning language-informed visual concepts by simply distilling pretrained vision-language models.
2. At inference time, the trained concept encoders extract concept embeddings from a test image, which can be remixed to generate images with novel compositions of concepts.
3. Using a light-weight test-time finetuning procedure, these encoders can also be quickly adapted to extract novel concepts *unseen* during training.

4. Experiments show that this visual concept representation achieves better disentanglement and compositionality, compared to text-based prompting baselines, as shown in Figures 2 and 6.

## 2 RELATED WORK

### 2.1 VISUAL CONCEPT LEARNING

Designing learning-based systems to discover various visual concepts in natural images has been a long-standing goal in machine perception and intelligence. Early attempts typically rely on extensive semantic annotations done by humans, such as object classification (Barnard et al., 2003; Fei-Fei et al., 2006; Fergus et al., 2005), which were later epitomized by the effort of ImageNet (Russakovsky et al., 2015). Visual concepts are intrinsically linked to concepts in language, and such end-to-end supervised learning paradigms can be seen as learning a direct mapping between visual concepts and discrete linguistic concepts. Other approaches attempt to better exploit this inherent structure in language by constructing a structured representation of visual concepts such as scene graphs (Zhong et al., 2021) and symbolic programs (Mao et al., 2019; Han et al., 2019).

More recently, the success of natural language modeling (Devlin et al., 2018; Brown et al., 2020; Raffel et al., 2020) has paved the way for grounding visual concepts to open vocabularies, unlike category labels or fixed symbolic programs, by training large Vision-Language Models (VLMs) on massive image captioning datasets (Schuhmann et al., 2022). This has powered recent Text-to-Image (T2I) generation models to turn linguistic concepts from free-form text prompts into photo-realistic images (Rombach et al., 2022; Saharia et al., 2022). These T2I models have been leveraged by Personalization methods for extracting individual visual concepts from one or a few images. This is done by either by optimizing token embeddings (Gal et al., 2022; Vinker et al., 2023; Avrahami et al., 2023; Chefer et al., 2023b; Liu et al., 2023), finetuning the backbone denoiser (Ruiz et al., 2023), or training additional encoders for amortized optimization (Gal et al., 2023; Arar et al., 2023; Li et al., 2023a). We also distill visual concepts from a pre-trained T2I model, but unlike existing works, we train encoders to adhere to a set of language-specified concept axes, preserving the disentangled and compositional nature of language. Ranasinghe & Ryoo (2023) also explores language-defined concepts but focuses on video action recognition tasks while we focus on image generation.

A separate line of work focuses on unsupervised visual concept disentanglement without explicitly leveraging language, typically by simply imposing information constraints in the latent space of a generative model, like VAEs and GANs (Higgins et al., 2017; Chen et al., 2016; Hsu et al., 2023). Here, we are interested in learning visual concepts that are explicitly grounded to language.

### 2.2 CONTROLLABLE IMAGE GENERATION

The success of GAN-based image generation (Goodfellow et al., 2014; Brock et al., 2018; Karras et al., 2019) has spawned a series of works that discover controllable directions in the GAN latent space (Voynov & Babenko, 2020; Härkönen et al., 2020). More recently, the advancements of diffusion-based T2I models have unlocked new possibilities for controllable image generation, where photo-realistic images can be generated from free-form text prompts. Recent works proposed to further improve the alignment of image samples and input text conditions by manipulating attention maps within T2I models (Chefer et al., 2023a; Epstein et al., 2023). Another form of controllable image generation is compositional generation. Liu et al. (2022) proposes to improve the quality of T2I diffusion models for composing multiple pre-given concepts, specified via text prompts, by modifying the inference procedure. In this work, instead of assuming that concepts are given and are in a text format, we tackle the task of identifying disentangled concepts which can be used for composition.

Image generation can also be controlled with image analogies (Šubrtová et al., 2023; Hertzmann et al., 2001), a form of visual prompting. These works typically do not explicitly extracts visual concepts from inputs unlike ours. In this work, we amalgamate both visual prompts and text queries, employing them as the editing interface.

## 3 METHOD

Fig. 3 gives an overview of our proposed learning framework. Our goal in this work is to extract visual concepts from images along a number of *concept axes* specified by language, such as

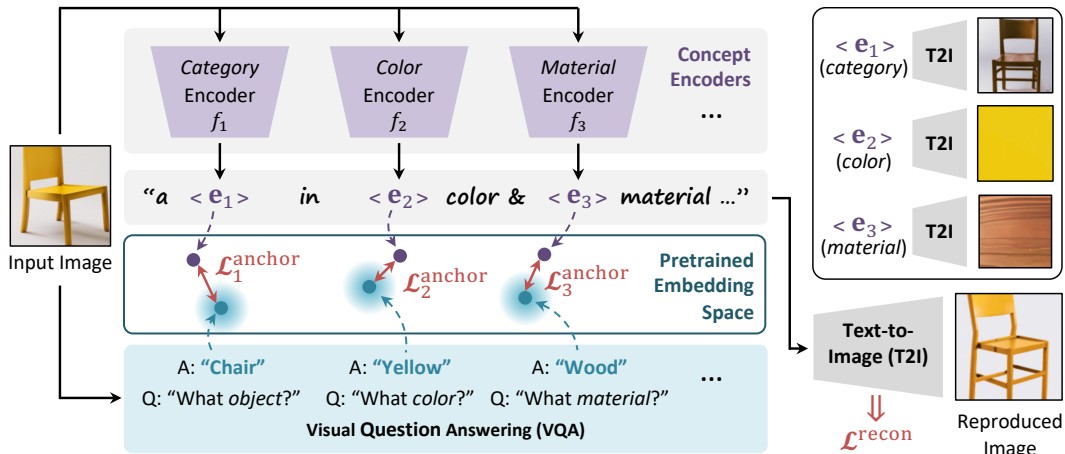

Figure 3: **Training Pipeline.** During training, an input image is processed by a set of concept encoders that predict concept embeddings specific to given concept axes. These embeddings are trained to (1) retain information in order to reproduce visual inputs via a pre-trained Text-to-Image model given an axis-informed text template, and (2) ensure disentanglement across different axes by anchoring to text embeddings obtained from a pre-trained Visual Question Answering model.

category, color, and material, so as to enable the flexible composition of concepts into high-quality image generations.

To achieve this, we train a set of visual concept encoders by distilling concept guidance from pre-trained vision-language models. Specifically, the encoders are trained to extract concept embeddings from an image in order to fulfill two objectives. First, they should be recomposed to explain the input image through a pretrained text-to-image (T2I) generation model, given a concept-axis-informed text prompt. Second, these visual concept embeddings should be anchored to the corresponding text embeddings obtained from a pre-trained visual question answering (VQA) model, further exploiting the disentangled nature of linguistic concepts for better disentanglement of visual concepts.

## 3.1 Visual Concept Encoding by Inverting Text-to-Image Generation

Our understanding of the visual world is centered around various concept axes, to which we have often assigned words due to their significance in communication and reasoning. This vision-language grounding has fueled recent explosion of text-to-image generation models (Rombach et al., 2022; Saharia et al., 2022; Ramesh et al., 2022), allowing them to generate photo-realistic images with various combinations of concepts defined by words.

Here, we are interested in the reverse direction of text-to-image generation, where the goal is to extract language-grounded visual concepts present in natural images. Specifically, given $K$ concept axes of interest defined by language, we would like to learn $K$ concept encoders $\{f_k(\cdot)\}_{k=1}^K$, each of which extracts a concept representation $\mathbf{e}_k = f_k(\mathbf{x})$ along a concept axis from an input image $\mathbf{x}$.

In order to train these concept encoders $\{f_k(\cdot)\}$, instead of relying on extensive human labeling, we opt to exploit the vision-language grounding embedded within large pre-trained T2I generation models. Using the technique of Textual Inversion (Gal et al., 2022), one can optimize a token embedding <∗> to capture a visual entity in a given image, through the objective of regenerating the image with the T2I model from a text template, such as "a photo of <∗>". Here, we adopt a similar objective, but instead of inverting a specific embedding capturing the overall "identity" of an individual image instance, we would like to predict embeddings $\mathbf{e}_k$ that are grounded to a number of meaningful concept axes, using an axis-informed text template, such as "a photo of <$\mathbf{e}_1$> with <$\mathbf{e}_2$> color and <$\mathbf{e}_3$> material". This allows the extracted concept embeddings to be shared across different images, encapsulating the common visual characteristics pertinent to one concept axis.

Specifically, given an image $\mathbf{x}$, the concept encoders $\{f_k(\cdot)\}$ extract a set of concept embeddings $\{\mathbf{e}_k \in \mathbb{R}^D\}$, which have the same dimension $D$ as the text embeddings so that they can be directly inserted into the text embeddings of the axis-informed text template. To simplify the notations, let $f_\gamma(\cdot)$ denote the function that takes in the image and produces the final sequence of embeddings

of the template and the predicted concept embeddings, and $\gamma$ be the parameters of all the encoders which will be optimized during training. Let $\mathbf{c}_\theta$ be the part of the T2I model's text encoder that takes in a sequence of text embeddings and outputs a conditioning vector for the T2I model's denoising network $\hat{\epsilon}_\theta$, where $\theta$ denotes network parameters. We use DeepFloyd (StabilityAI; Saharia et al., 2022) as the backbone T2I model, which utilizes a pre-trained T5 model (Raffel et al., 2020) as the text encoder, and keep the parameters $\theta$ frozen in all experiments. To train the encoders, we reuse the training objective for the backbone diffusion model:

$$\mathcal{L}^{\text{recon}}(\mathbf{x}; \gamma) = \mathbb{E}_{\epsilon \sim \mathcal{N}(\mathbf{0}, I), t \sim \mathcal{U}([0,1])} \left[ \|\hat{\epsilon}_\theta(\mathbf{x}, t, \mathbf{c}_\theta(f_\gamma(\mathbf{x}))) - \epsilon\|_2^2 \right], \tag{1}$$

where the noise $\epsilon$ is sampled from a standardmultivariate Gaussian distribution and the timestep $t$ is sampled from a uniform distribution in $[0, 1]$. Minimizing $\mathcal{L}^{\text{recon}}$ amounts to finding concept embeddings within the space of the pre-trained T2I model that can best reproduce the input image $\mathbf{x}$, resembling a "reconstrucion" objective.

Compared to per-instance token optimization in vanilla Textual Inversion, the advantages of training these concept encoders are two-fold. First, the concept embedding space is naturally shared across different image instances, encapsulating the common understanding of the corresponding concept axes. Second, it makes training more efficient by amortizing the optimization across all instances, and more crucially, it allows for test-time inference in a feed-forward pass.

### 3.2 CONCEPT DISENTANGLEMENT USING TEXT ANCHORS

The objective of $\mathcal{L}^{\text{recon}}$ ensures that the extracted concept embeddings can sufficiently reconstruct the concept of a given image through a pre-trained text-to-image generation model. However, with this loss alone, there is little guarantee that each embedding encodes *only* the information pertinent to a particular concept axis. In practice, we found that this baseline results in poor disentanglement of different concept axes when remixing the concept embeddings to generate new images, potentially due to the imprecise vision-language grounding in the pre-trained T2I model. For instance, as shown in Figure 8, the extracted `category` embedding `<e₁>` for "red berries" cannot be remixed with various `color` embeddings `<e₂>` *e.g.*, "orange", as `<e₁>` is highly entangled with the concept of a "red" color due to the bias in natural images.

To encourage better disentanglement of different concept axes, we further incorporate a sparse set of text anchors into the concept embedding space. Along each concept axis like `color`, we have often named some prominent modes, such as "red" or "yellow", and these text labels entail clearly disentangled concepts. Therefore, we would like to reuse this disentangled nature of linguistic concepts to improve the disentanglement of visual concepts. To this end, we make use of the text predictions from a pre-trained Visual Question Answering (VQA) model, BLIP-2 (Li et al., 2023b), as pseudo ground-truth anchors for the concept embeddings.

Specifically, for each training image $\mathbf{x}$ and for each concept axis of interest (*e.g.*, `color`) indexed by $k$, we query the BLIP-2 model $\Psi$ with the image $\mathbf{x}$ and a question $q_k$ in natural language that is specific to this concept axis, *e.g.*, "what is the `color` of the object in the image". Denote the answer from BLIP-2, also in the form of natural language, as $\Psi(\mathbf{x}, q_k)$. We encode this answer with the pre-trained text encoder $c_\theta$ to obtain a text embedding $\tilde{\mathbf{e}}_k = c_\theta(\Psi(\mathbf{x}, q_k))$. The prediction of our concept encoders $f_{k,\gamma}$ is encouraged to stay close to this anchor text embedding:

$$\mathcal{L}_k^{\text{anchor}}(\mathbf{x}; \gamma) = \|f_{k,\gamma}(\mathbf{x}) - \tilde{\mathbf{e}}_k\|_2^2, \quad \text{where} \quad \tilde{\mathbf{e}}_k = c_\theta(\Psi(\mathbf{x}, q_k)). \tag{2}$$

It is crucial to highlight that we use these BLIP-2 predictions *only as anchors* by assigning a small weight to this anchor loss $\mathcal{L}_k^{\text{anchor}}$ during training. Otherwise, the embeddings predicted by the concept encoders could easily collapse to a set of discrete text embeddings and fail to capture the visual nuances in images.

### 3.3 TRAINING AND INFERENCE

**Training.** Given a collection of training images $\mathcal{D}$ containing various combinations of concepts along each axis, the final objective to train the concept encoders consists of the two parts:

$$\mathcal{L}^{\text{total}}(\gamma) = \mathbb{E}_{\mathbf{x} \sim \mathcal{D}} \left[ \mathcal{L}^{\text{recon}}(\mathbf{x}; \gamma) + \sum_{k=1}^{K} \lambda_k \mathcal{L}_k^{\text{anchor}}(\mathbf{x}; \gamma) \right]. \tag{3}$$

**Inference.**  At inference time, given a new test image, the concept encoders extract embeddings $\{\mathbf{e}_k\}$ capturing its characteristics along each concept axis of interest. These embeddings can be remixed across different images, or be replaced by embeddings converted from explicit words, to produce images with new compositions of visual concepts through the backbone T2I generator.

**Generalization to Unseen Concepts via Test-Time Finetuning.**  While the encoders can precisely extract an axis-specific concept that has been seen during training from a new test image, they tend to be less robust to concepts unseen at training. However, with a lightweight test-time optimization procedure, where we use only the reconstruction objective $\mathcal{L}^{\text{recon}}$ to update the parameters for all encoders, $\gamma$, these encoders can generalize to novel concepts unseen during training. Note that $\mathcal{L}^{\text{anchor}}$ is omitted here in order to capture the visual nuances without over-committing to the coarse text anchors. After training, the encoders have learned to generate outputs within a relatively narrow region of the embedding space, which allows the model to adapt to the test images shown in Figure 5 within around 600 iterations while maintaining disentanglement and compositional capability.

## 4 EXPERIMENTS

### 4.1 EXPERIMENT SETUP

**Training Data Generation.**  We train the concept encoders only using synthetic images generated by DeepFloyd from 5 different domains, including *fruits*, *figurines*, *furniture*, *art*, and *clothing*. More details of our dataset can be found in A.2. For each dataset, we consider 2-3 concept axes, such as `category`, `color`, `material`, `style`, and `season`. For example, considering `category` and `color` for the *fruits* dataset, we generate training images by prompting DeepFloyd with text prompts describing varying combinations of categories and colors, *e.g.* "a photo of an apple which is red in color". Note that these text prompts are used only for data generation and not for training, as they may not be reliable (Figure 2). On average, we obtain 669 training images for each dataset.

**Implementation Details.**  Inspired by Gal et al. (2023), we leverage a pre-trained CLIP ViT/L-14 model for image encoding (Radford et al., 2021; Dosovitskiy et al., 2021), which was trained with a contrastive objective aligning image and text features, and hence well-suited for our task. We extract image features from CLIP ViT and train $K$ separate concept encoders $f_k$ on top of the features, which share the same architecture but maintains separate weights. Specifically, we take the `[CLS]` tokens from each CLIP layer and process each token with a distinct linear layer. This is different from Gal et al. (2023), which extracts `[CLS]` tokens from every other layer and uses a single shared linear layer for all token features. The transformed features are then aggregated with average pooling followed by a LeakyReLU (Xu et al., 2015) activation, and passed into another linear layer that produces the final predicted concept embeddings.

To ground the concept embeddings to the concept axes, we adapt the text templates from CLIP (Radford et al., 2021), which were originally used to assemble captions with class categories from ImageNet (Russakovsky et al., 2015). For training, we use AdamW (Loshchilov & Hutter, 2017) optimizer with learning rate 0.02, and randomly flip the images horizontally. For test-time finetuning, we use the AdamW optimizer with learning rate 0.001. We set $\lambda_k = 0.0001$ (Equation (3)) for the `category` axis and $\lambda = 0.001$ for others. We use IF-I-XL from DeepFloyd as the backbone model, with training resolution $64 \times 64$. Training on one dataset takes approximately 12 hours on one NVIDIA GeForce RTX 3090 GPU. Generated images are upsampled $256 \times 256$ using IF-II-L for visualization purpose.

### 4.2 QUALITATIVE RESULTS

**Visual Concept Extraction, Recomposition and Extrapolation.**  Once trained, the concept encoders can extract *disentangled* concept embeddings specific to each concept axis from different test images, which can recomposed to generate new images with various concept compositions. As shown in Figure 4, across various datasets, our method is able to recompose axis-specific visual concepts from different images and consistently generate new images depicting the recomposed concepts. More examples can be found in Appendix A.1, where images generated from individual decomposed concept embeddings are also presented.

This disentangled concept representation also allows us to *extrapolate* along a particular concept axis for visual exploration, as shown in Figure 7. For instance, we can ask BLIP-2 "what is the style

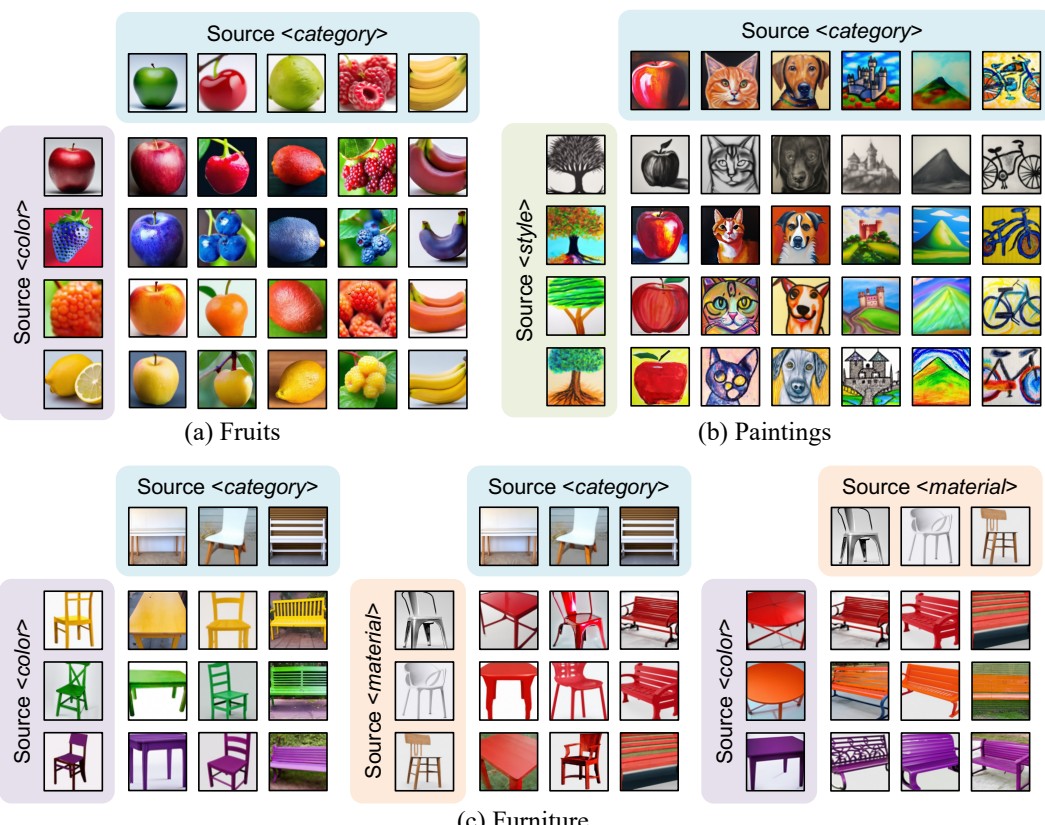

Figure 4: **Concept Recomposition.** At test time, our model extracts visual concepts along various axes from different images and recompose them to generate new images. We show recomposition results across different pairs of concept axes in 3 datasets: (a) *Fruits*, (b) *Paintings*, and (c) *Furniture*.

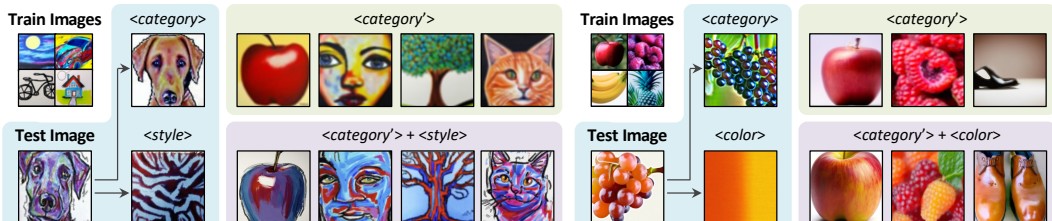

Figure 5: **Generalization to Unseen Concepts via Finetuning.** After test-time fine-tuning on a single test-time image, encoders can adapt to novel concept. Visual details from the input images are preserved as can be seen from images visualizing embedding predictions. Importantly, these embeddings do not overfit to the input images and maintain a good disentanglement, such that they can be freely recomposed to create new concepts. More results can be found in Figures 9 to 13. More real-world results can be found in Figures 20 to 24.

of the painting?" in an image, and prompt GPT-4 (OpenAI, 2023) to name a few alternatives. We can then recompose the text embeddings of these alternative styles with our concept embeddings and generate images to visualize these variants. Representing concepts as continuous embeddings further enables concept interpolations. Details and results are shown in Appendix A.6.

**Generalization to Unseen Concepts via Test-Time Finetuning.** Although the encoders have only seen a limited range of concepts during training due to the small size of the training dataset, it can be quickly adapted to unseen concepts with the lightweight test-time finetuning procedure in Section 3.3, as shown in Figure 5. For instance, after 600 finetuning iterations, the model can extract the specific style of the dog painting unseen at training and compose it with the content from other images. It can also capture nuanced colors *e.g.* yellow-ish-orange and transfer them to other objects.

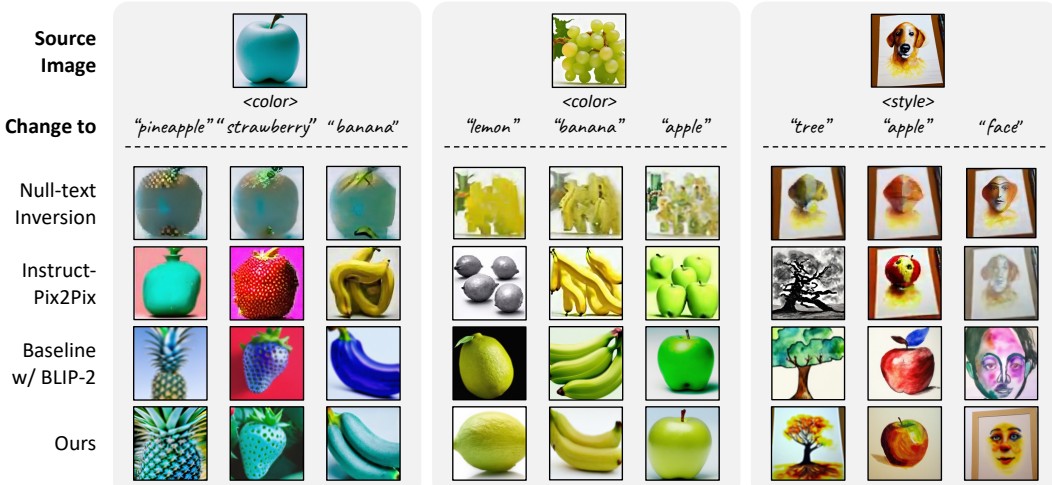

Figure 6: **Baselines Comparisons.** Compared the text-based image editing methods, our method achieves significantly better compositionality due to the disentangled concept representation, and captures fine-grained color variations, which the baseline struggles to encode with language.

| | CLIP-Score ↑ | | | | | | Human Evaluation ↑ | |
| | Edit Category | | | Edit Color | | | Edit Cat. | Edit Clr. |
| | Cat.&Clr. | Cat. | Clr. | Cat.&Clr. | Cat. | Clr. | Score | Score |
|---|---|---|---|---|---|---|---|---|
| Null-text Inversion | 0.258 | 0.249 | **0.249** | 0.259 | 0.265 | 0.223 | 0.287 | 0.316 |
| InstructPix2Pix | 0.267 | 0.277 | 0.226 | 0.270 | 0.245 | **0.268** | 0.233 | 0.648 |
| Baseline w/ BLIP-2 | **0.313** | 0.294 | 0.248 | 0.287 | 0.271 | 0.237 | 0.448 | 0.379 |
| Ours | 0.308 | 0.297 | 0.238 | **0.302** | **0.287** | 0.236 | **0.968** | **0.840** |
| w/o $\mathcal{L}_k^{\text{anchor}}$ | 0.268 | 0.276 | 0.219 | 0.263 | 0.257 | 0.236 | - | - |
| w/o Encoder & $\mathcal{L}_k^{\text{anchor}}$ | 0.288 | **0.300** | 0.214 | 0.242 | 0.213 | 0.265 | - | - |

Table 1: **Quantitative Comparisons on Visual Concept Editing.** Compared to existing image editing baselines (Brooks et al., 2023; Mokady et al., 2022), our method achieves better overall CLIP score when editing either axis, and is particularly effective at retaining `category`-related concepts as reflected in human evaluation. 'Cat' denotes Category and 'Clr' denotes Color.

## 4.3 Comparison with Prior Works

**Qualitative Comparisons** While this task of image generation with disentangled visual concepts is new, we identified prior work that is capable of text-based image editing and generation, and establish a side-by-side comparison on the task of *visual concept editing*. Specifically, given an image $\mathbf{x}$, for example, of a teal-colored apple in Figure 6, the task is to generate a new image $\hat{\mathbf{x}}_{\mathbf{e}_k \to \mathbf{e}_k'}$ with one concept axis $k$ (*e.g.* `category`) modified by a text prompt, from $<\mathbf{e}_k>$ to $<\mathbf{e}_k'>$, while preserving other axes $\{i | i \neq k\}$ from the input.

We compare to two existing methods. Null-text Inversion (Mokady et al., 2022) performs concept editing by first inverting the input image to token embeddings of Stable-Diffusion (Saharia et al., 2022) and then applying Prompt-to-Prompt (Hertz et al., 2022), which modifies cross attention maps for editing, a process that leads to pixel-aligned editing results. InstructPix2Pix (Brooks et al., 2023) is a conditional diffusion model for text-based image editing. Since it is trained on data curated with Prompt-to-Prompt for supervision, it also tends to generate pixel-aligned results. We also design a naive baseline for this task, where we query BLIP-2 for a text description of the attributes to be retained from the test image, and combine the answer with a target concept to generate the final recomposed image. As shown in Figure 6, our method achieves better recomposition results, whereas baseline methods fail to disentangle the desired axis-specific concepts from input images, and therefore struggle to faithfully reproduce the desired concepts from the image and the text.

**Quantitative Comparisons.** We also conduct quantitative evaluations on the task of text-based visual concept editing and compare with prior work. Specifically, we record the ground-truth text

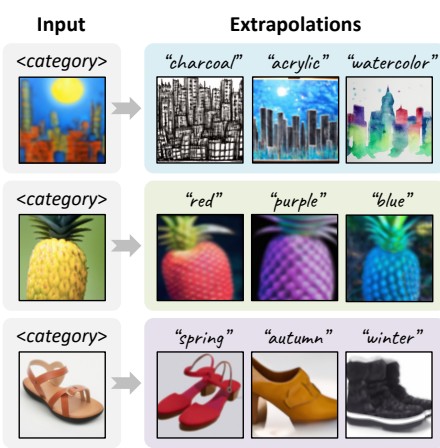

Figure 7: **Visual Concept Extrapolation**. Given an input image, we can extrapolate along a concept axis by querying BLIP-2 and GPT-4 to name a few alternatives to the concept in the input. Our model can then generate new variants of the input for visual exploration, by mixing them with the extracted concept embeddings.

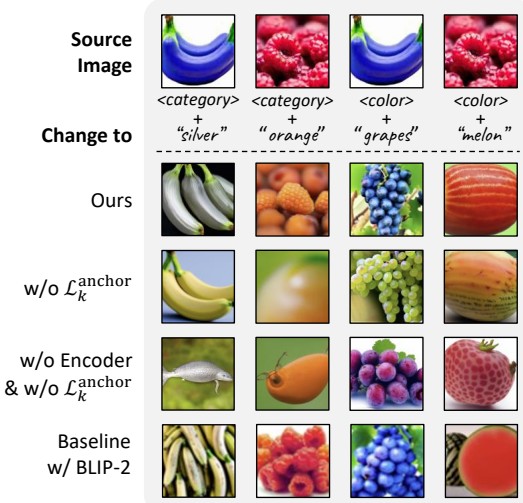

Figure 8: **Ablations.** Both the text anchor loss and the encoder design reduce overfitting to input images and improve concept disentanglement. Removing these components significantly deteriorates the visual concept editing results. Additionally, we observe that the BLIP-2 baseline has difficulties transferring unnatural colors.

prompts $y$ that we used to generate each training image $\mathbf{x}$, and manually filter out the inconsistent pairs. For each concept axis $k$, we randomly select a target label $<\mathbf{e}_k'>$ from a set of candidates that different from the one in the original prompt $<\mathbf{e}_k>$. We then measure the alignment score between the modified image $\hat{\mathbf{x}}_{\mathbf{e}_k \to \mathbf{e}_k'}$ with the modified text prompt $y_{\mathbf{e}_k \to \mathbf{e}_k'}$ using CLIP (Radford et al., 2021), following Gal et al. (2022). We also break down the score to each individual axis, by computing the alignment of the image to a prompt specifying *only one* axis, *e.g.* "a photo of $<\mathbf{e}_k'>$". All results are summarized in Table 1. Our method captures the characteristics of category particularly well and outperforms others in changing category or preserving category while changing color, which highlights its ability to extract disentangled concepts allowing for flexible compositions. Instruct-Pix2Pix is more effective in changing color, but tends to do poorly in preserving the category, as seen in Figure 6. More details on our quantitative results can be found in A.4.

We further conduct a human evaluation. Each participant is presented the results from all three methods in a random order together with the editing instruction, and asked to rank them considering both realism and faithfulness to the instruction (see Appendix A.3 for details). We aggregate the responses from 20 people and report the average average score normalized to 0-1 in Table 1. More details of our setup can be found in A.3.

### 4.4 ABLATIONS

We conduct an ablation study to understand the effects of the proposed encoder (as opposed per-instance optimization) and the text anchoring loss $\mathcal{L}_k^{\text{anchor}}$. We use the same evaluation dataset and the CLIP-alignment metric as described in Section 4.3. As shown in Fig. 8 and Table 1, removing $\mathcal{L}_k^{\text{anchor}}$ and the encoders deteriorates disentanglement of the different concept axes due to severe overfitting, resulting in poor recomposition results. More results are included in the Appendix.

## 5 CONCLUSION

In this paper, we have presented a framework for learning language-informed visual concepts from images, by simply distilling from pre-trained vision-language models. After training, the concept encoders extract *disentangled* concept embeddings along various concept axes specified by language, which can be remixed or edited to generate images with novel concept compositions. We conduct thorough evaluations both quantitatively and qualitatively, and demonstrate that our approach yields superior results in visual concept editing compared to prior work.

## ACKNOWLEDGMENTS

We thank Kyle Hsu, Joy Hsu, and Stephen Tian for their detailed comments and feedback. This work is in part supported by NSF RI #2211258 and #2338203, ONR MURI N00014-22-1-2740, ONR N00014-23-1-2355, AFOSR YIP FA9550-23-1-0127, the Stanford Institute for Human-Centered AI (HAI), Amazon, J.P. Morgan, and Samsung.

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

# A APPENDIX

## A.1 MORE QUALITATIVE RESULTS

More qualitative results for concept extraction and recomposition are shown in Figures 9 to 13. Across various datasets, our method achieves superior disentanglement and recomposition results.

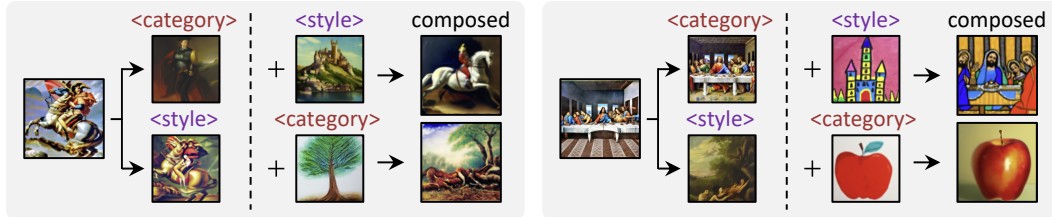

Figure 9: Visual Concept Extraction and Recomposition Results on *Art* dataset.

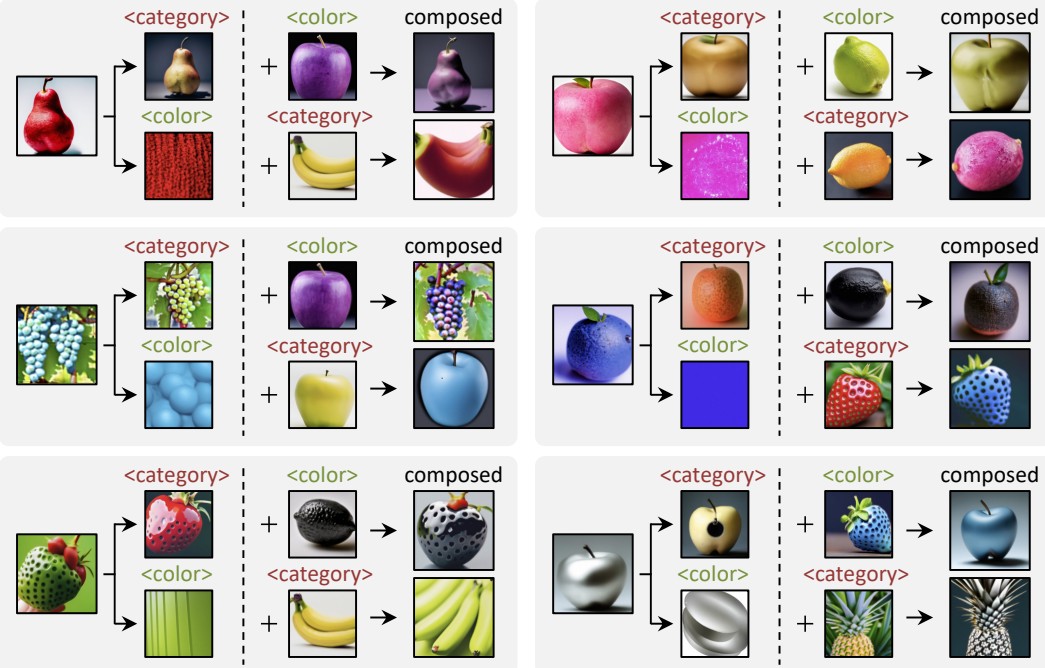

Figure 10: Visual Concept Extraction and Recomposition Results on *Fruits* dataset.

## A.2 DATASETS

We use 5 training datasets for the experiments. The concept axes in interest, together with ground truth words used in the text prompt for DeepFloyd to generate the training data, are listed below.

*Fruit* Dataset.
`Category`: Cherry, Apple, Banana, Mango, Strawberry, Pineapple, Lemon, Raspberry
`Color`: Red, Blue, Green, Purple, Black, Yellow, Orange

*Art* Dataset.
`Category`: Beach, Tree, Apple, Cat, Dog, Face, City, Hill, Sky, Car, Bike, House, Castle, Chair
`Style`: Charcoal, Oil, Paint, Acrylic, Crayon, Pastel

*Figurine* Dataset.
`Category`: Ball, Block, Statue
`Color`: Red, Blue, Green, Yellow, White, Cream, Purple, Pink
`Material`: Plastic, Fur, Wood, Metal, Leather

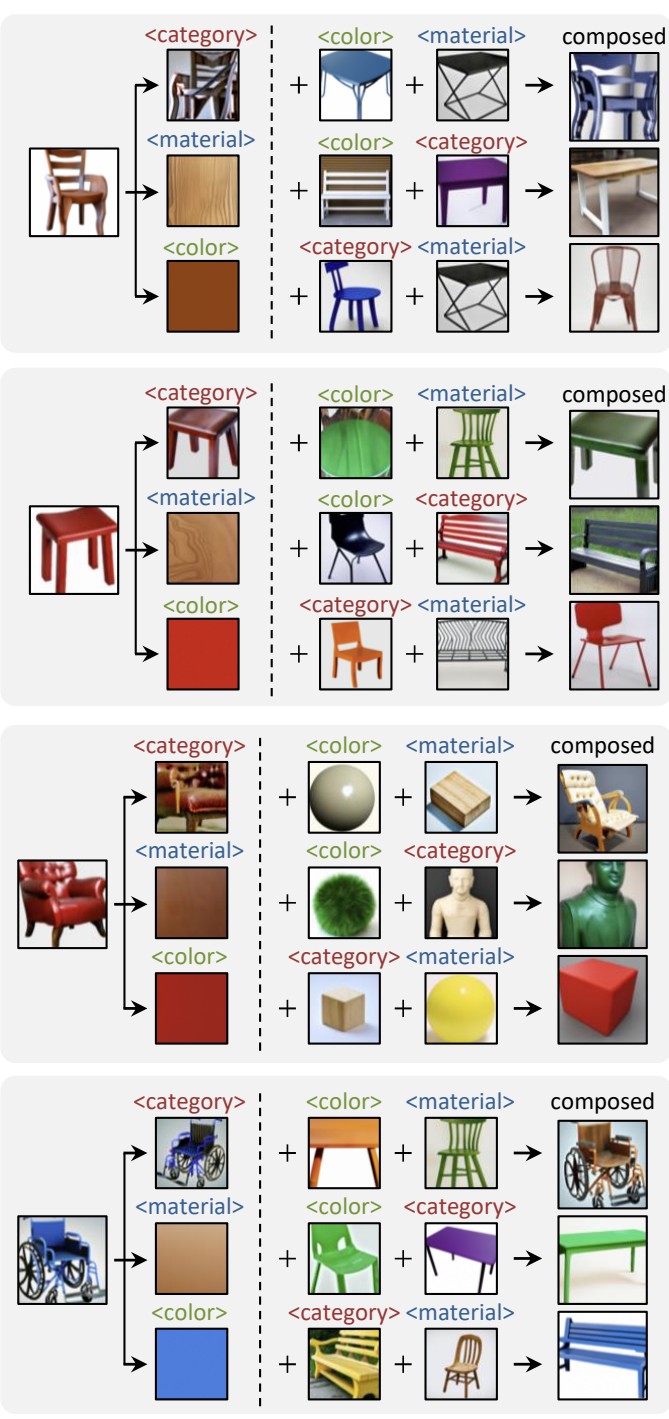

Figure 11: Visual Concept Extraction and Recomposition Results on *Figurine* dataset.

*Clothing* Dataset.
Category: Shirt, Pants, Shoes, Dress, Cap
Color: Red, Yellow, Green, Purple, White, Cream
Season: Spring, Summer, Fall, Winter

*Furniture* Dataset.
Category: Chair, Table, Bench

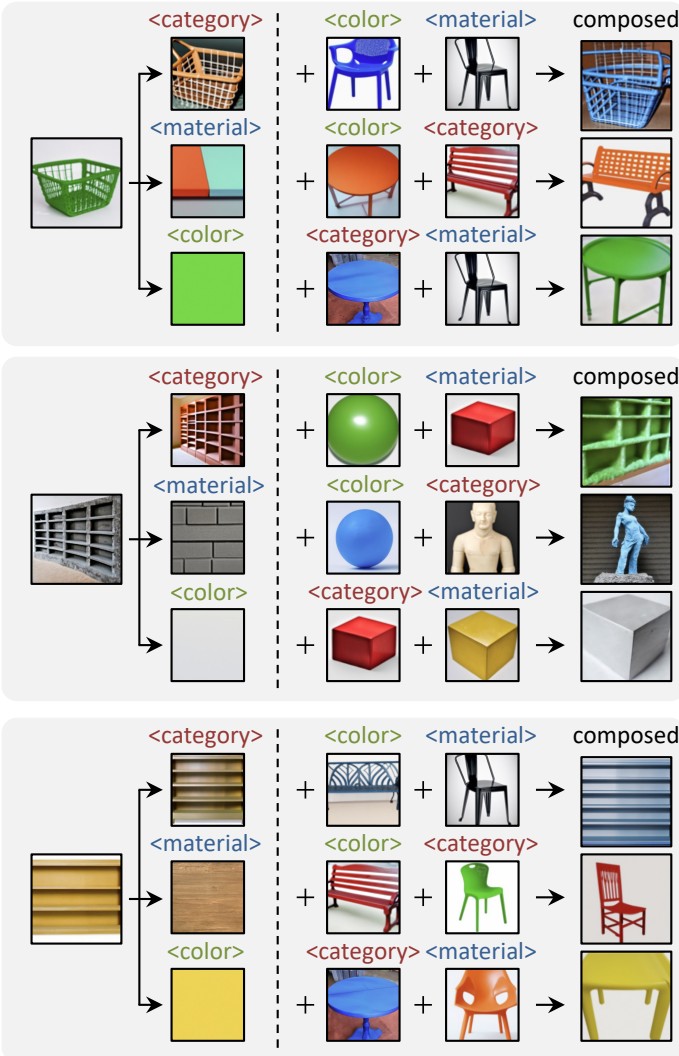

Figure 12: Visual Concept Extraction and Recomposition Results on *Objects* dataset.

Color: Red, Orange, Yellow, Green, Blue, Purple, Black, White
Material: Wood, Metal, Plastic

### A.3 HUMAN EVALUATION

In the user study, we create a questionnaire where users are asked to choose the image that matches our test setting: given an image prompt and text instruction, users rank images that best represents both the image prompt and text instruction. Here, we randomly sample 12 data points from the evaluation dataset for color editing and 12 for category editing. We collected the questionnaire responses from 20 users. Our score follows the Borda point metric (Saari, 2023) where 1st, 2nd, and 3rd would receive 2, 1, and 0 point(s) respectively, allowing us to differentiate rankings of 3, 2, 2 and 2, 1, 1. We then calculate the average scores over both the number of questions and participants, and subsequently normalize these values to a $0 - 1$ scale. Results are shown in 1.

A set of instructions Figure 14 are presented to the respondents before the user study. Then, they are presented with the 24 questions of color and category editing tasks. Example questions are shown in Figures 15 and 16.

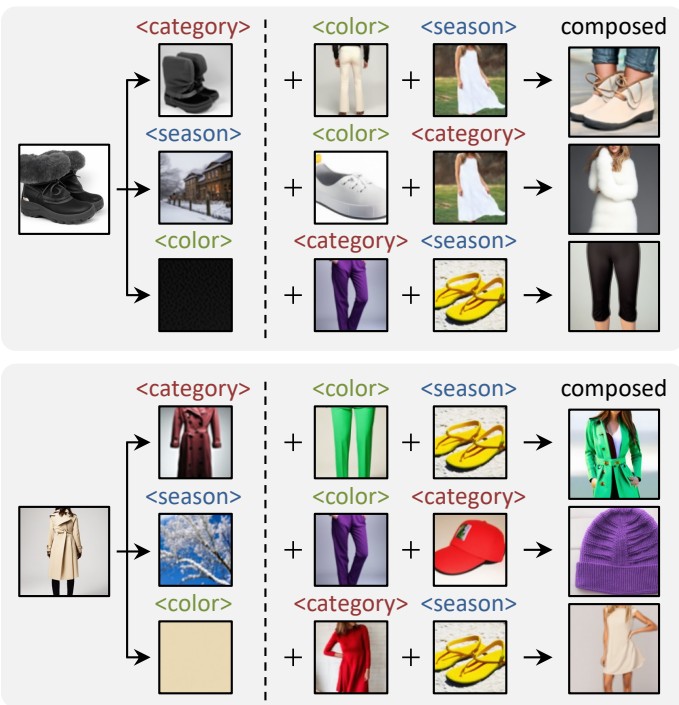

Figure 13: Visual Concept Extraction and Recomposition Results on *Clothing* dataset.

In this survey, you will be presented with pairs of images alongside a specific caption that describes what the images should depict. Your task is to evaluate and select the image that most closely adheres to the following criteria:

🌍 Fidelity: The image should appear realistic, with convincing colors, textures, and details.
🍎 Caption Adherence: The image should accurately represent the content described in the caption. For example, if the caption states "a red apple," the fruit should both be "red" and represent the shape/ texture of an "apple".
🍇 Your ranking should reflect the specific shade of the sample image's color, but not the specific instance of the sample image's fruit. 1 is the best, while 3 is the worst.
💪 If none of the answers seem accurate, answer with your best guess.

Additional Guidelines: Take your time to inspect each image carefully before making your selection. Images can be tied e.g. we can have an order of

- 1, 1, 2
- 1, 2, 2
- 1, 2, 3

Figure 14: **Questionnaire Instructions.**

### A.4 QUANTITATIVE ANALYSIS.

Figure 17 analyzes the mean and standard deviation of CLIP scores introduced in Section 4.3. Our method achieves the best score with a low variance for both the category (left) and the overall (right) metric, and stays competitive on the color (metric).

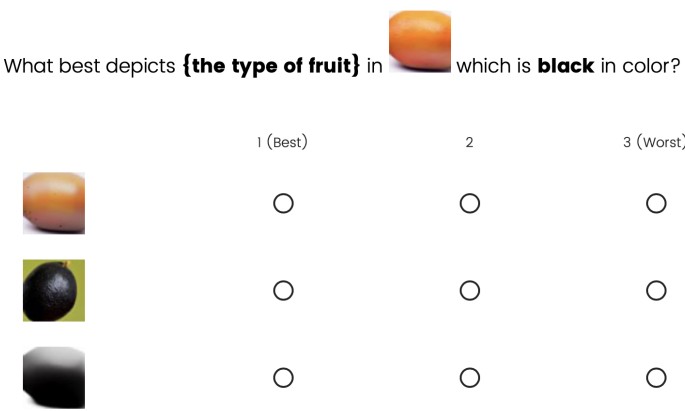

Figure 15: **Questionnaire Attribute Modification.**

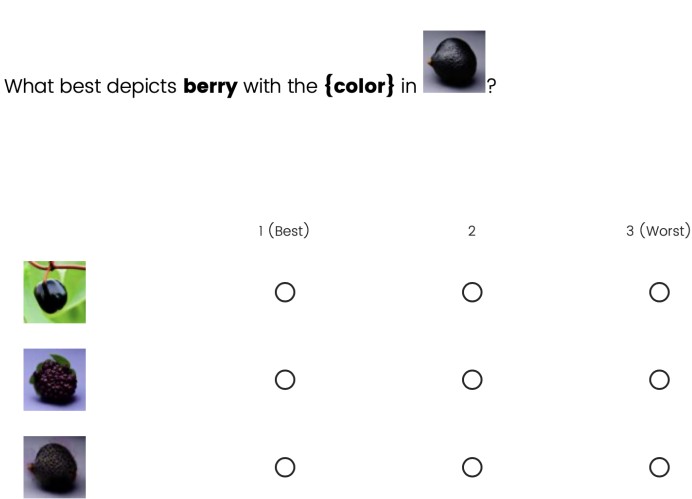

Figure 16: **Questionnaire Category Modification.**

## A.5 DATASET LABELLING WITH BLIP

In our approach, we employ BLIP in order to annotate each image with its respective category and attributes. To achieve this, we query BLIP for each image, with one question for each attribute or category. For example, for an image of a red wooden table, we would ask BLIP what is the name of the item, what material it is made of, and what color it is.

During the pre-training stage, items corresponding to identical prompts and therefore the same place-holder tokens are aggregated by taking the most commonly occurring BLIP answer for each category or attribute. For example, if we have 8 images of red wooden tables, and one of them is misclassified as a red plastic table, we would still label it with 'red', 'wood', and 'table' for color, material, and category respectively.

## A.6 INTERPOLATION OF CONCEPT EMBEDDINGS

We further demonstrate concept interpolation results. By interpolating between two concept embeddings, our model can generate meaningful images depicting gradual changes from one concept to another, such as the hybrid fruit of cherries and bananas shown in Figure 18. To interpolate between two input images, first, we extract CLIP features for both images, giving us two vectors of size $12 \times 1024$. We interpolate the two vectors using Spherical interpolation (SLERP). Specif-

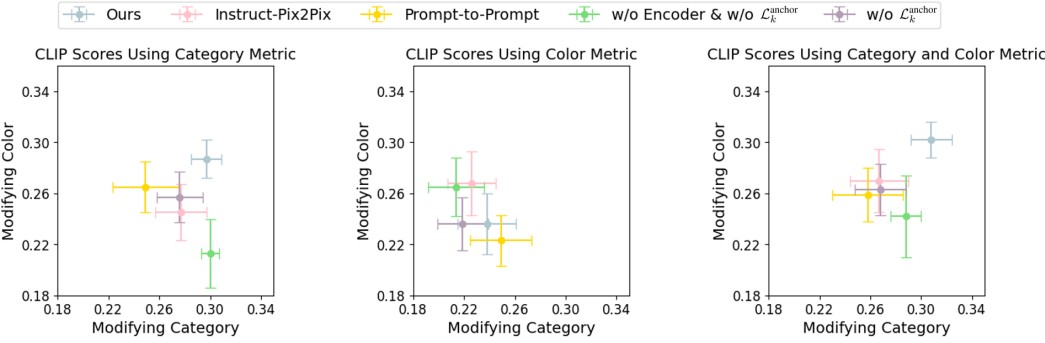

Figure 17: **Quantitative Baselines.**

ically, given two normalized vectors $A_{norm}$ and $B_{norm}$ of dimensions $12 \times 1024$, we compute the dot product to find the cosine of the angle $\theta$ between them as $\cos(\theta) = A_{norm} \cdot B_{norm}$. For 12 interpolation points, each point $i$ is calculated using $\alpha_i = \frac{i}{11}$ and the interpolated vector is $slerp(A, B, \alpha_i) = \frac{\sin((1-\alpha_i)\theta)}{\sin(\theta)} A_{norm} + \frac{\sin(\alpha_i\theta)}{\sin(\theta)} B_{norm}$. This ensures the resultant vectors maintain a constant magnitude.

With a set of trained encoders, each encoder takes in the interpolated CLIP features, and their outputs are combined with a training-time textual template to generate the interpolation results as shown in Figures 18 and 19.

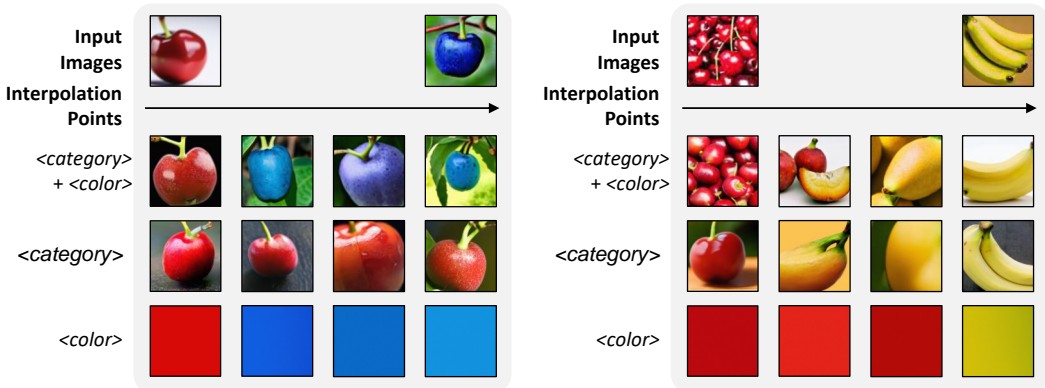

Figure 18: Interpolation on *Fruit* dataset.

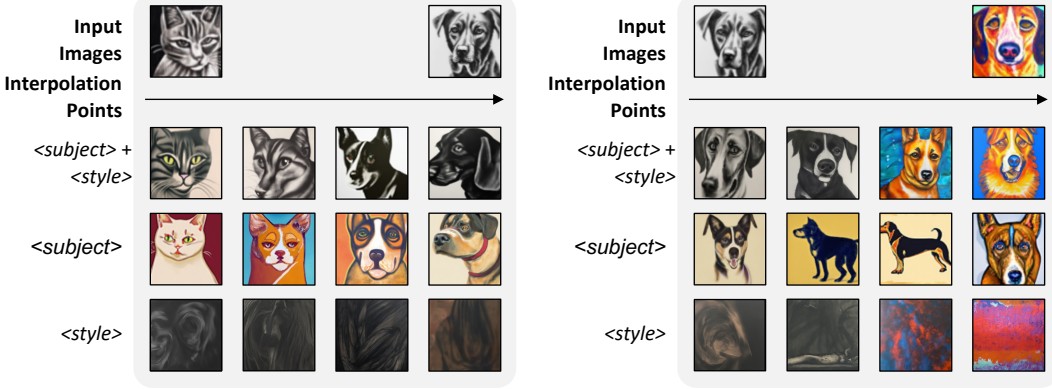

Figure 19: Interpolation on *Art* dataset.

## B  INFERENCE ON REAL-WORLD IMAGES

Despite being trained on images generated by diffusion-based models, the concept encoders generalize well to diverse, complex real-world images, including *unseen* types of object category, material, and color from casual photo captures (Figures 20 to 23) and unseen types of artwork style (Figures 5 and 24).

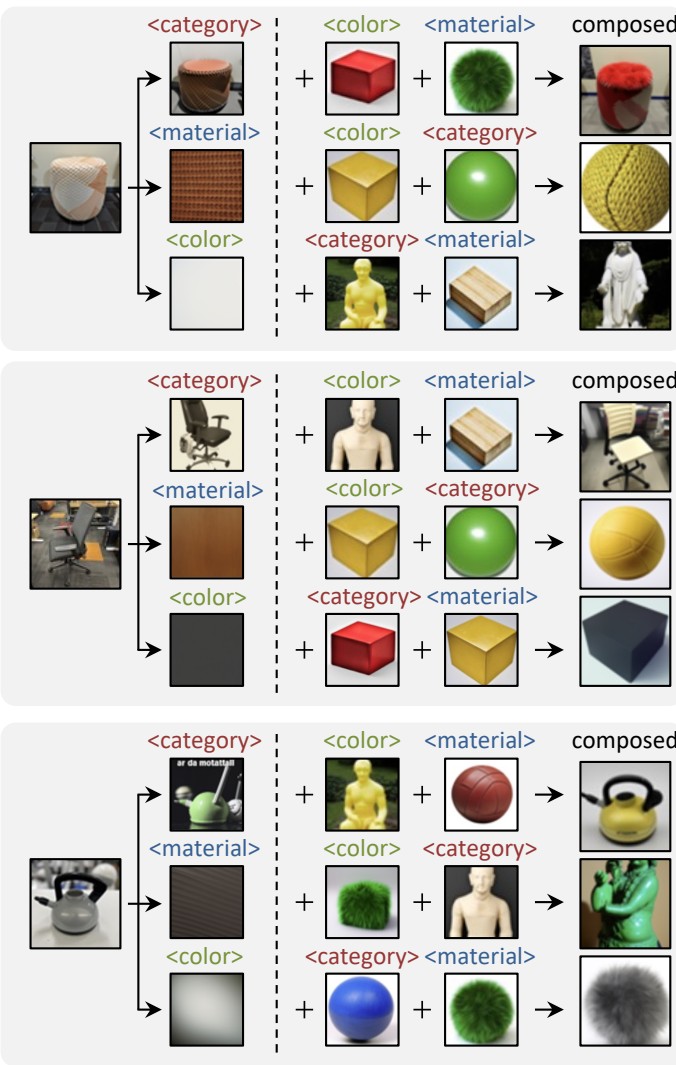

Figure 20: Visual Concept Extraction and Recomposition Results on Real-world Images of various objects.

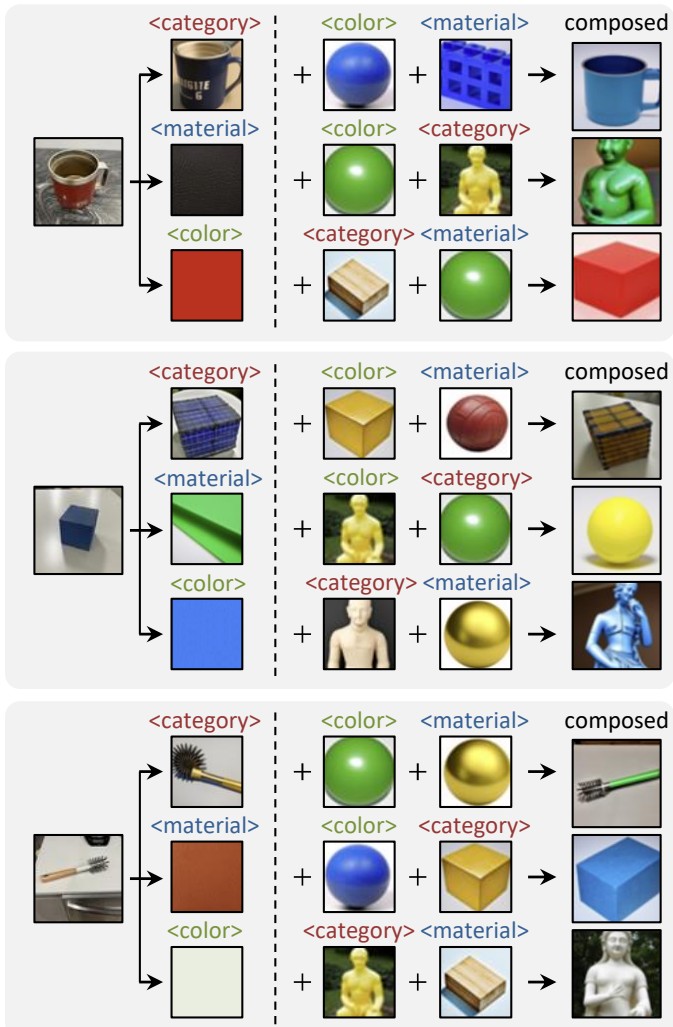

Figure 21: Additional Visual Concept Extraction and Recomposition Results on Real-world Images of various objects.

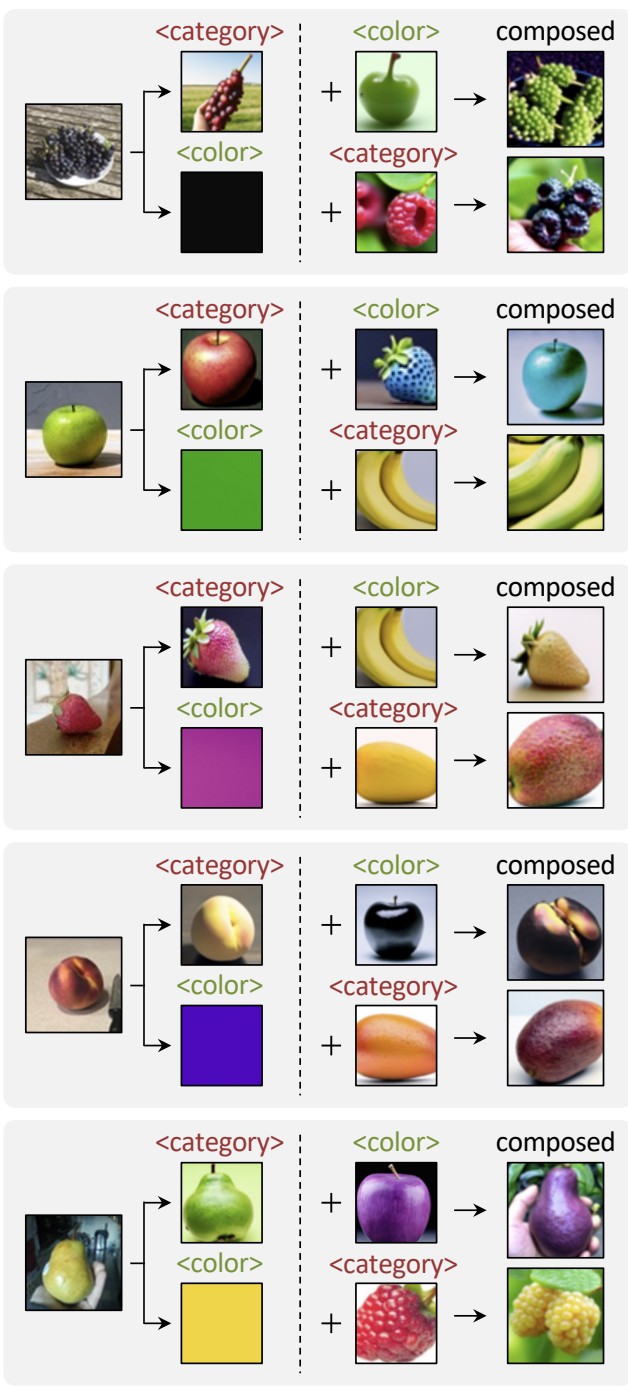

Figure 22: Visual Concept Extraction and Recomposition Results on Real-world Images of various fruits.

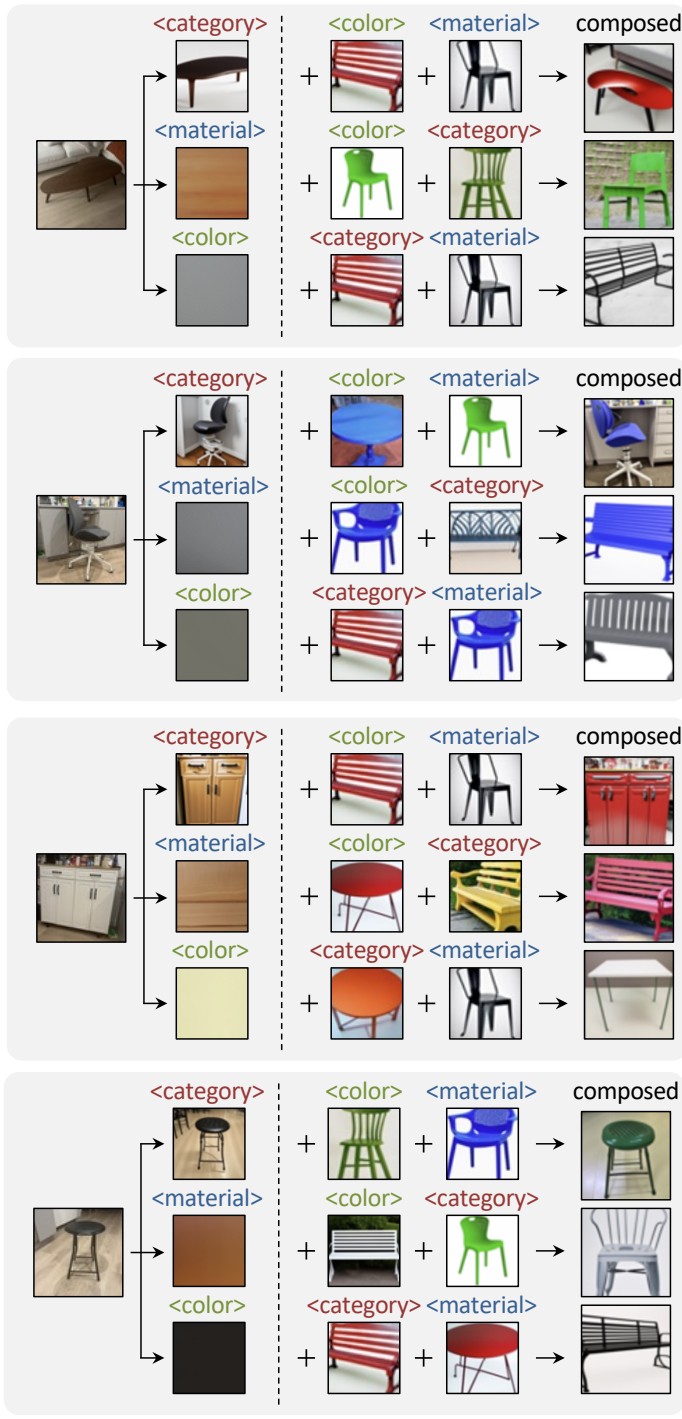

Figure 23: Visual Concept Extraction and Recomposition Results on Real-world Images of various pieces of furniture.

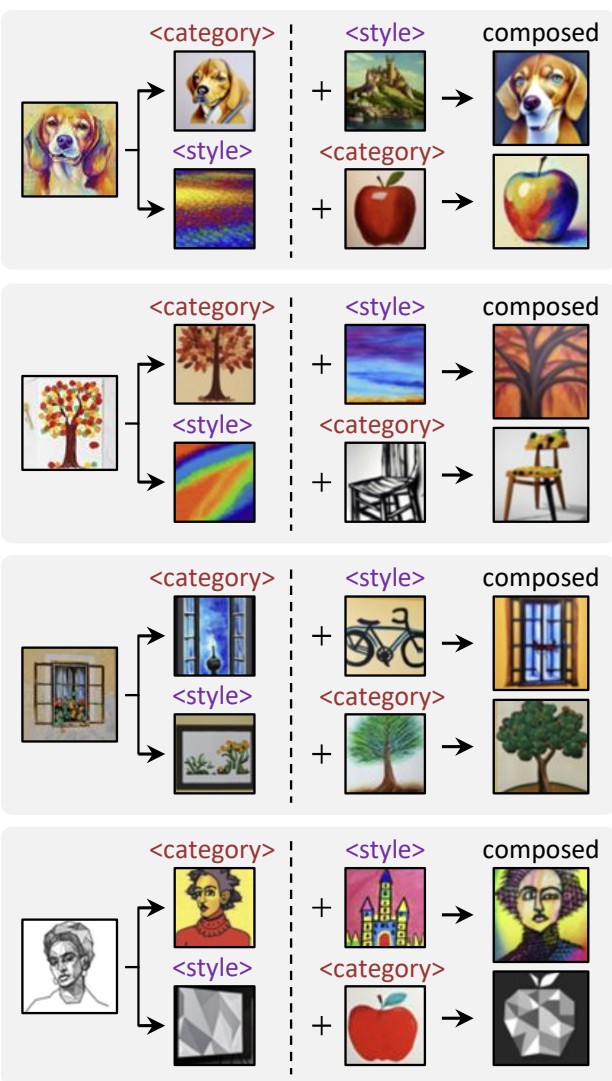

Figure 24: Visual Concept Extraction and Recomposition Results on Real-world Images of artwork.

## C THE EFFECT OF THE ANCHOR LOSS

During training time, the anchor loss (Equation (2)) encourages the encoder predictions to converge to a *meaningful* subspace within the word embedding space Gal et al. (2023). This ensures that these embeddings can be readily visualized by a pre-trained text-to-image generation model, and improves the compositionality across different concept axes, as shown in Figure 6.

Empirically, we find that simply setting a small weight on this loss can effectively achieve this objective, allowing the model to capture nuances along each axis, without collapsing to the word embeddings. In Figure 25, we empirically show such examples, where we compare the concept embeddings predicted by the color encoder to the text embedding of the training-time BLIP-2 label, *e.g.* "blue" from Figure 25, and the former preserves the specific color of the input image while the latter does not.

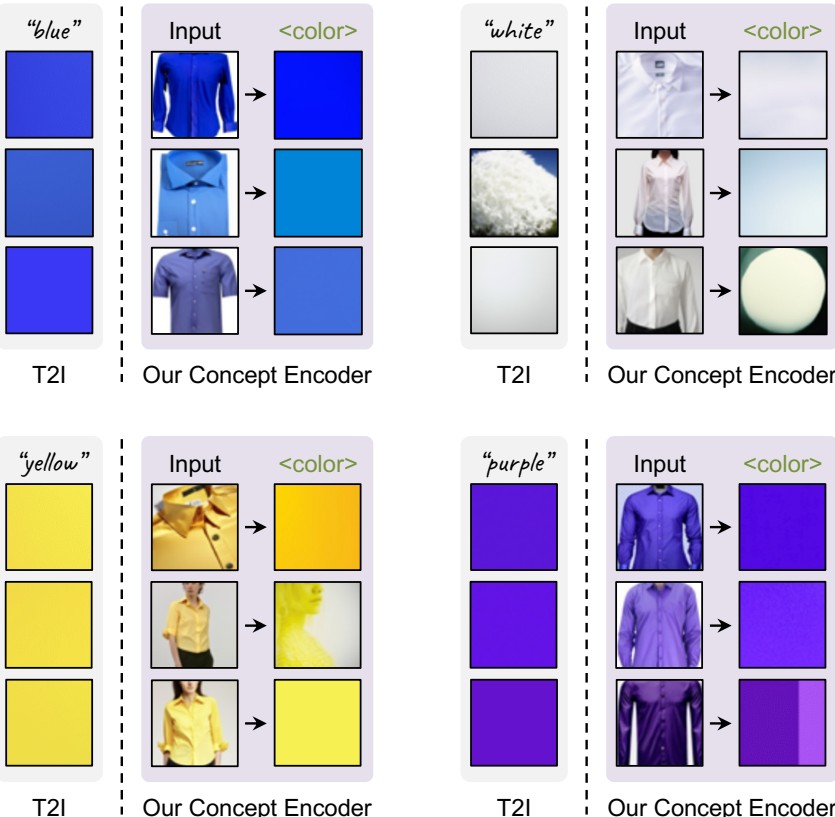

Figure 25: The concept embeddings extracted by our concept encoders capture various shades of colors instead of generic 'blue', 'white' *etc*. directly generated by the T2I model DeepFloyd.

## D EXTENSION WITH NEW CONCEPT AXES

We show that we are able to extend to additional concept axes by training new concept encoders *without retraining the existing ones*. In the experiments shown in Figures 26 and 27, given two trained encoders for category and color and the corresponding training dataset, we train the material encoder while keeping the other two frozen. We show that such procedural training maintains the disentanglement of the frozen encoders while allowing the framework to extend to a new concept axis.

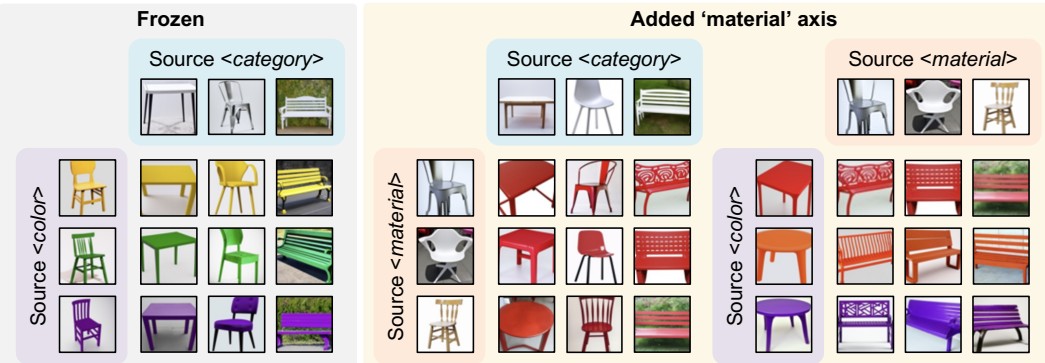

Figure 26: Concept Recomposition after adding a new material axis to the model trained with category and color axes.

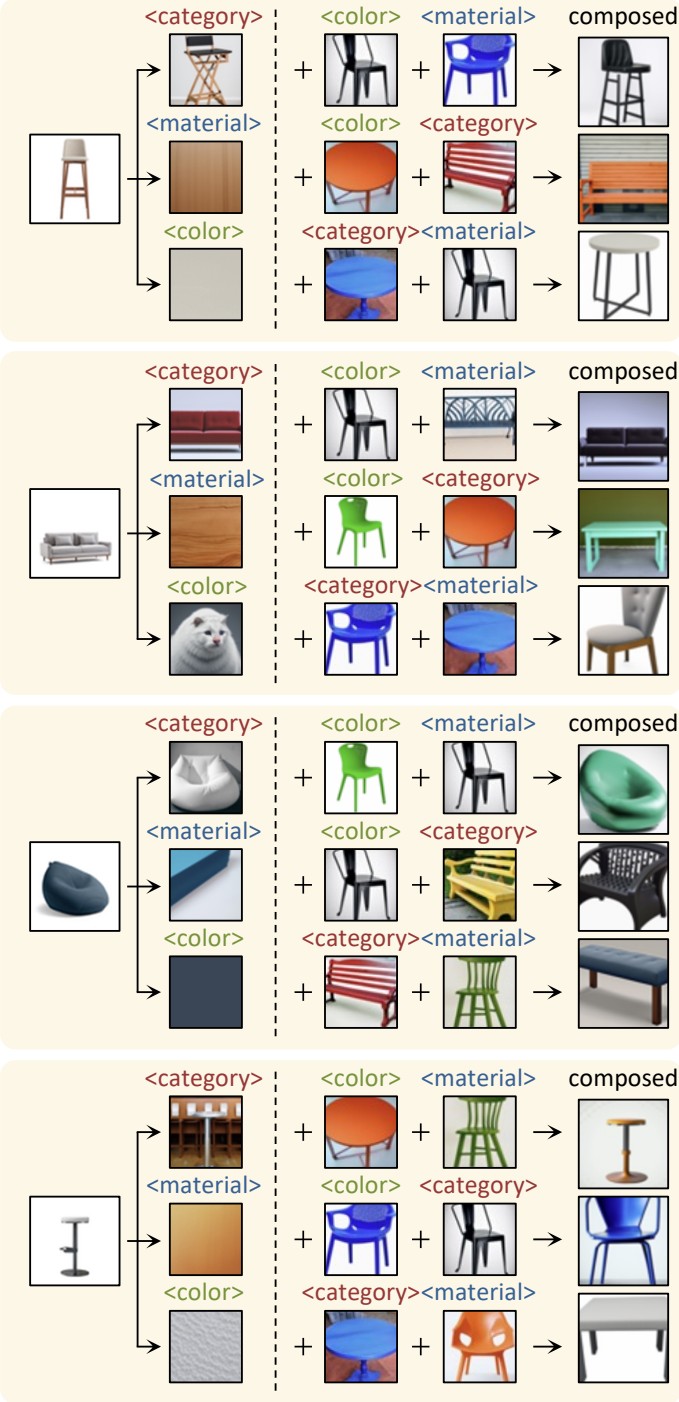

Figure 27: Test-Time Generalization Results on images of furniture, where the `material` encoder is additionally trained on top of the frozen `category` and `color` encoders.

# E    ADDITIONAL BASELINE COMPARISONS

In Figure 28, we provide additional qualitative examples accompanying the experiments in Table 1. From these visual examples, we observed that the color nuances captured by ours are more accurate compared to the BLIP-2 baseline. However, since the CLIP-based metric specified in Section 4.3 cannot fully capture the minute differences, the BLIP-2 baseline still achieves comparable scores to our method despite this evident gap in visual results.

To quantify such visual differences in colors, we compare the color of a $16 \times 16$ centered patch from the input image and the image generated by the method being evaluated, both of resolution $64 \times 64$, and report the L2 error of the mean color of the two patches. We report the average of metric across all examples in Figure 28 in Table 2. Results suggest that our method captures colors more accurately compared to the baselines.

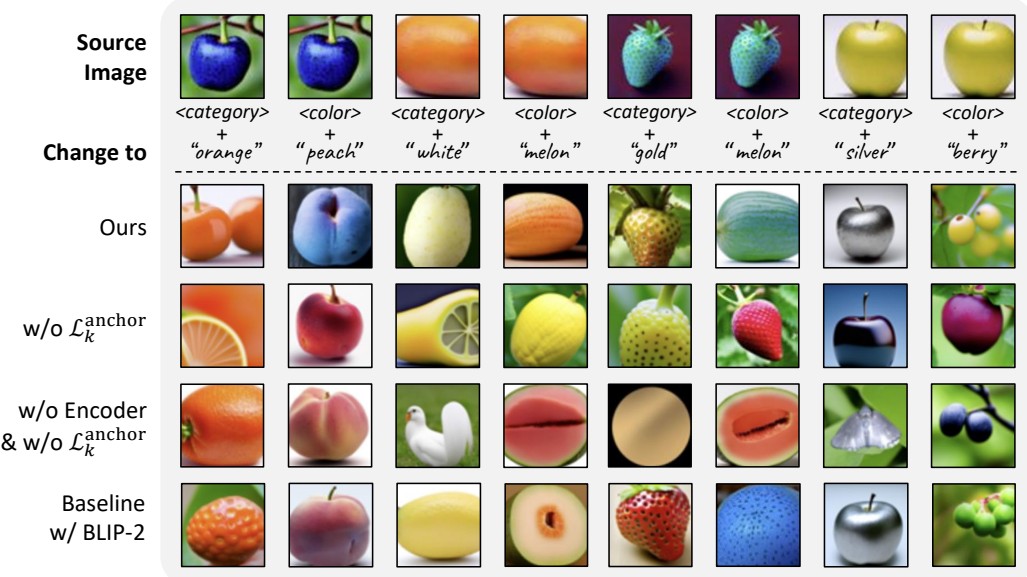

Figure 28: More qualitative results for ablations.

|  | CIELAB $\Delta$E* $\downarrow$ | | | |
|---|---|---|---|---|
|  | Cherry | Mango | Strawberry | Apple |
| Ours | **35.50** | **4.76** | **16.12** | **15.64** |
| w/o $\mathcal{L}_k^{\text{anchor}}$ | 101.34 | 86.34 | 85.31 | 122.01 |
| w/o Encoder & $\mathcal{L}_k^{\text{anchor}}$ | 85.86 | 82.46 | 82.07 | 127.80 |
| Baseline w/ BLIP-2 | 79.90 | 25.20 | 47.30 | 73.62 |

Table 2: **Quantitative Comparisons on Color Editing.** To quantify the differences in color seen in Figure 28, we use CIELAB $\Delta$E*, the color-distance metric recommended by the International Commision on Illumination (Fraser et al., 2004).

# F  DISCUSSION ON TEXTUAL INVERSION

As discussed in Section 3.1, compared to directly using text as inputs for image generation, using techniques like Textual Inversion (Gal et al., 2022), our model is able to capture more nuanced visual details of a particular image with continuous embeddings, instead of discrete words. This can be illustrated in the empirical results in Figures 6, 8 and 28, which show that our method preserves the nuances from input images more accurately than the BLIP-2 baseline which uses texts for conditional generation.

## F.1  LIMITATIONS

Given a few concept axes specified by language, the current model is able to learn disentangled concept embeddings for various concept remixing and editing tasks. However, this requires that the concept axes are given a priori to the model, which limits the generality of the concept space that can be captured by the model. Moreover, we train separate encoders for each concept axis, which does not make full use of the inherent hierarchical structure of these concept axes.

