# OpenReview forum: "Language-Informed Visual Concept Learning"
_ICLR.cc/2024/Conference — ICLR 2024 poster_

### Official Review · Reviewer_jNDi · 2023-10-29

**Soundness:** 3 good
**Presentation:** 3 good
**Contribution:** 2 fair
**Rating:** 6
**Confidence:** 3

**Summary:**

The paper presents a method for learning concepts by distilling knowledge in a text-to-image generative model. The method assumes concept axes, in each of which specific information of an input image is encoded (like colors, materials, and object categories). The method learns a designated encoder for each concept axis with generated images. For training the encoders, the method uses an anchor loss for each axis based on a VQA model to further disentangle the axes and a reconstruction loss. The method is evaluated qualitatively and quantitatively (CLIPScore and human evaluation).

**Strengths:**

(1) The paper is well-written. I can easily follow what is done in the method.

(2) The method is simple and can be trained in 12 hours only with less than 1000 generated images, yet outperforming the similar existing methods.

(3) The image generation results are really nice compared to the existing approaches.

**Weaknesses:**

(1) The performance of the method is mostly shown in qualitative evaluations. The quantitative evaluation only shows the performance of image generation by modifying some concept axes (and human evaluation). I think the paper would be better if it came with an objective quantitative evaluation of the obtained concepts themselves in some ways (though I didn’t come up with any good approaches for this).

(2) Related to (1), I’m not sure if CLIPScore is really sensitive to arbitrary combinations of concepts. Some references or experimental results may help understand the experiment.

(3) The paper’s purpose is not sufficiently clear. Is it to learn concepts for image generation? Or is it for some other downstream tasks?

**Questions:**

I would like to have some responses for (1)-(3) in the weakness section.

---

> ### Author Response · Authors · 2023-11-23
> **Response for Reviewer jNDi**
>
> **Q1 - Additional quantitative evaluation**
>
> To better quantify the performance on capturing nuanced colors, we designed a new metric based on color statistics for the task of concept editing as an additional quantitative evaluation.
>
> Specifically, we compare the color of a 16 × 16 patch cropped around the center (focusing on the object) from both the input image and the image generated by the method being evaluated. The original size of the image is  64 × 64. We then compute the MSE of the mean color of the two patches, and report the average of the metric across all examples in Figure 11 in Table 1. Results suggest that our method captures colors more accurately compared to the baselines.
>
>
> **Q2 - Additional Qualitative results accompanying the CLIP-score evaluation**
>
> In addition to the results in the main paper, in Figure 11 of the `Rebuttal PDF`, we provide more qualitative examples accompanying the experiments in Table 1. From these visual examples, we observed that the color nuances captured by ours are more accurate compared to the BLIP-2 baseline. However, since the CLIP-based metric specified in Section 4.3 of the paper cannot fully capture the minute differences, the BLIP-2 baseline still  achieves comparable scores to our method despite this evident gap in visual results. To better quantify such visual differences in colors, we further designed a new metric based on color statistics, as explained in Q3 above.
>
>
> **Q3 - Purpose of the framework**
>
> The goal of visual concept learning is to extract visual representations in a structured manner akin to human perception. Humans interpret and organize the visual world into a hierarchy of abstract concepts, such as `object categories`, `colors`, `styles` etc. With this abstracted visual concept representation, we can easily picture a *new* visual instance with various concept compositions, such as `a red banana`.
>
> In this paper, we aim at developing a method that can automatically learn to extract disentangled image representations along a number of language-specified concept axes (`category`, `color`, etc).
> With these disentangled representations, we can then recompose them to *generate new compositions of visual concepts*, capturing visual nuances which would otherwise often exceed the limitations of language.

---

### Official Review · Reviewer_Bcg9 · 2023-10-30

**Soundness:** 2 fair
**Presentation:** 3 good
**Contribution:** 2 fair
**Rating:** 5
**Confidence:** 5

**Summary:**

The authors claim that their proposed model can learn a language-informed visual concept representation, by simply distilling large pre-trained vision-language models.

**Strengths:**

The authors claim that their proposed model can learn a language-informed visual concept representation, by simply distilling large pre-trained vision-language models.

**Weaknesses:**

1. What is concept representation learning? Is concept learning just the mutual translation of text and images?

2. In the experiments, the authors primarily focus on conducting investigations using synthetic datasets. However, it raises concerns about the generalizability of the conclusions/findings obtained from synthetic datasets to real-world datasets.

3. The concept learning should focus more on the understanding of concepts, especially at different granularities of the same concept.

**Questions:**

Please refer to Weakness.

---

> ### Author Response · Authors · 2023-11-23
> **Response for Reviewer Bcg9**
>
> **Q1 - What is visual concept representation learning?**
>
> The goal of visual concept learning is to extract visual representations in a structured manner akin to human perception. Humans interpret and organize the visual world into a hierarchy of abstract concepts, such as object `categories`, `colors`, `styles` etc. With this abstracted visual concept representation, we can easily picture a *new* visual instance with various concept compositions, such as `a red banana`.
>
> In this paper, we aim at developing a method that can automatically learn to extract disentangled image representations along a number of language-specified concept axes (`category`, `color`, etc).
> With these disentangled representations, we can then recompose them to *generate new compositions of visual concepts*, capturing visual nuances which would otherwise often exceed the limitations of language.
>
>
> **Q2 - Real-world datasets**
>
> We provide new inference results on real-world images in Figures 1 to 5 from the Rebuttal PDF, spanning various types of objects, including furniture, kitchen utensils, fruits, and artwork. Despite *only* being trained on images generated by diffusion-based models, the concept encoders generalize well to diverse, complex real-world images, including *unseen* types of objects, materials, colors, and styles. Note that some of the examples originally presented in the main paper also came from real photos, such as Figure 5 from the main paper.

---

### Official Review · Reviewer_dyyk · 2023-10-31

**Soundness:** 2 fair
**Presentation:** 2 fair
**Contribution:** 3 good
**Rating:** 5
**Confidence:** 4

**Summary:**

This paper proposed a new framework for Visual Concept Learning. By introducing a set of concept encoders, concept embeddings could be extracted from the input image, which could be recomposed later to produce desired image. The experiments showed that these concept embeddings could capture the visual nuances and they are disentangled with each other. Besides, this framework can learn the shared concepts across instances(images), in other words, it is more efficient than previous methods.

The paper proposes a new framework for VCL task that avoids massive human annotation, and could boost the research in related fields. It also drew the attention to the research direction of using continuous embeddings (instead of relying on generic words) as the visual concept descriptors. The concept is commendable, although the depiction falls short of perfection and would greatly benefit from additional elaboration and intricate explanations.

**Strengths:**

a. Proposed the idea to use BLIP-2 generated embeddings as “anchor” (pseudo ground-truth embeddings) to “separate” the entangled concept embeddings.

b. This work proposed a framework to tackle Visual Concept Learning without human annotation, which is much more efficient than the previous works.

c. Unlike most of the Textual Inversion techniques, this work could capture the concept appearing in different images. Therefore, it does not require a retraining on each image. Also, the learning efficiency is higher because it can learn from a larger pool of images.

**Weaknesses:**

a. The datasets used for the experiment were small and simple. It is not guaranteed that the claimed conclusions could be maintained when this framework is applied on more complex datasets (with much more concepts). The idea of using anchors to ensure that the embeddings are disentangle is great, however more experiments on larger datasets should be done to prove it. Given that there are only 2 to 3 concepts in each domain, the sparsity of concepts might be one of the reasons why the embeddings are disentangle.

b. The effectiveness of L^{anchor} is not fully explained. The L^{anchor} is omitted during the test-time optimization procedure to avoid “over-committing to the coarse text anchors”. However, in the ablation experiment, the paper claims “disabling L^{anchor} deteriorates the performance”. It seems kind of contradictory, the paper should explain more about why “disabling L^{anchor}” is desired during one phase but it leads to unsatisfactory results in general evaluation.

c. The ablation test is not fully explained. In “Editing Category” column, the results of “w/o Encoder & L^{anchor}_k” is actually higher than the results of “w/o L^{anchor}_k” in two metrics. This does not fully conform to the conclusion, quote, “and further removing the encoder decreases the overall decomposition performance”.

d. It is hard for this framework to generalize to new “concept”. From what I understood, this framework could effectively generalize to new “mode” of seen “concept” (like new style or new color), but not to new “concept”. When applied to new concepts, e.g. “size” or “shape”, the corresponding concept encoders need to be trained. Also, from my perspective, we can’t only train the concept encoder of the new concepts. Because the sentence template “a photo of <e1> with <e2> color and <e3>  and <e4>...” needs to cover at least the majority of the concepts appeared to generate an image close enough to the original input image. Based on this understanding, when this framework is extended to new concepts, the trained concept encoders (of the seen concepts) need to be retrained together with the new ones. This setting is not more efficient than previous methods.

**Questions:**

a. On page 5, in sentence “text encoder c_{theta} of the T2I diffusion model…”, it should be “part of the text encoder c_{theta}”. Because a text encoder should take “text” as input rather than “text embeddings”. The original sentence might be confusing.

b. On page 5, in formula (1), there is no explanation about the N and U notation. I assume they represent “multivariate normal distribution” and “uniform distribution” respectively. It would be more clear to annotate.

c. On page 5, in sentence “so that they can directly inserted into the text embeddings”, it seems that a “be” is missed.

d. The paper should mention the backbone T2I model earlier. It is first mentioned on page 5, it would be better to do it earlier.

e. It would be better if the choice of using “12 CLIP layers” over “6 out of 12 tokens like Gal et al. 2023” is explained more in detailed.

f. More details could be added about the test-time lightweight finetuning process.

---

> ### Author Response · Authors · 2023-11-23
> **Response for Reviewer dyyk**
>
> **Q1 - More complex datasets**
>
> It is worth highlighting that despite training on simple, synthetic datasets, the model generalizes well to diverse, complex *real-world images* as discussed in the general response above. Figures 1-5 show results on real-world photos of various types of objects, including furniture, kitchen utensils, fruits, and artwork. Despite the *unseen* types of objects, materials, and colors, the model still extracts disentangled and compositional concept embeddings, showing the strong generalization capability of the proposed framework to complex real-world images.
>
>
> **Q2 - Omitting the anchor loss during test-time optimization**
>
> During *training time*, $L^{anchor}$ encourages disentanglement of the concept embeddings across different concept axes while remaining in the subspace of the text embedding along their respective axes, as explained in Section 3.2. During *test time*, the concept encoders have already learned to encode only axis-specific concepts. The main objective is then to find a concept embedding within a specified axis that is consistent with the input image, which can be enforced by $L^\text{recon}$ (Equation (1)) alone. Omitting $L^{anchor}$ is beneficial as it helps the model to better preserve visual details of the input instance which are hard to capture with texts, *e.g.*, the style of paintings in Figure 24.
>
>
> **Q4 - Explanations on ablation study results**
>
> Thanks for pointing this out. In fact, both ablation baselines ‘w/o Encoder & $L^{anchor}_k$’ and ‘w/o $L^{anchor}_k$’ fail to learn disentangled concept embeddings, as illustrated in Figure 8 in the main paper as well as the additional results in Figure 11 of the `Rebuttal PDF`.
> The CLIP-based score is no longer indicative of the performance difference of the two baselines.
> We have carefully updated the conclusion in Section 4.4 in the paper.
>
>
> **Q5 - Extending with new concept axes**
>
> Our model can be extended with more concept axes by training new concept encoders *without retraining the existing ones*, as shown in Figures 9 and 10. Given a model trained with concept encoders for `category` and `color`, we additionally train a third encoder for `material` while keeping the other two encoders *frozen*. With this progressive training procedure, the model is able to extend to new concept axes while maintaining the disentanglement of the frozen encoders.
>
>
> **Q6 - Other clarifications**
>
> Thanks for pointing them out. We revised the paper accordingly, marked in blue.

---

### Official Review · Reviewer_qJGK · 2023-10-31

**Soundness:** 2 fair
**Presentation:** 3 good
**Contribution:** 2 fair
**Rating:** 8
**Confidence:** 3

**Summary:**

The authors use multiple visual encoders to disentangle various visual concepts from images. These visual concepts are defined as vector axes based on natural language description. The proposed framework performs a simple training on a synthetic dataset that learns disentangled vectors for each concept. These disentangled vectors can be combined with language (similar to textual inversion paper) to generate images containing combined concepts. They show how their method disentangles and generates new images with variably joined concepts better than existing work.

**Strengths:**

1. The paper explores an interesting idea of separating concepts in images in the visual domain in a novel manner
2. Interesting use of a VQA model to augment their training setup
3. Good use of diagrams to explain idea
4. Clearly showcase qualitative improvements for selected cases

**Weaknesses:**

1. The generality of method on real world images (i.e. where visual concepts are not that easily disentangled) is unclear
2. Limited evaluation (only one set of quantitative numbers)
3. Some missing details (refer questions below)

* Table 1: Are you reporting CLIP score and human evaluation on same table??
Please point out CLEARLY that these are two different metrics in the Table caption. Or please separate into two Tables. This is highly confusing.

---
The authors sufficiently respond to raised concerns in rebuttal.
1. The generality of the method (on real world images) is verified with multiple qualitative and quantitative evaluated added to appendix during rebuttal.
2. Additional quantitative evaluation with comparison to more baselines are presented.
3. The requested missing details are provided adequately.

Due to these reasons, I vote to accept this paper.

**Questions:**

* Immediate question - why don't concept axis vectors collapse to same as text embeddings? Explain this more.
* DeepFloyd - please cite appropriately
* Consider more discussion on Textual Inversion (as related work), maybe in supplementary at least. Highlight cases where this is better than directly using text.
* The work in [1] explores language defined concept axes in video domain - maybe an interesting comparison to discuss in related work
* Please include BLIP-based baseline results also in Table 1
* Can you add more CLIP-score based (or a different new metric based) evaluations for other task (like concept extrapolation)? More quantitative evaluation could really strengthen the paper

[1] Ranasinghe, K., & Ryoo, M., Language-based Action Concept Spaces Improve Video Self-Supervised Learning, NeurIPS 2023

---

> ### Author Response · Authors · 2023-11-23
> **Response for Reviewer qJGK**
>
> **Q1 - Real-world Images**
>
> We provide new inference results on real-world images in Figures 1 to 5 from the Rebuttal PDF, spanning various types of objects, including furniture, kitchen utensils, fruits, and artwork. Despite *only* being trained on images generated by diffusion-based models, the concept encoders generalize well to diverse, complex real-world images, including *unseen* types of objects, materials, colors, and styles. Note that some of the examples originally presented in the main paper also came from real photos, such as Figure 5 from the main paper.
>
>
>
> **Q2 - Additional quantitative evaluation**
>
> To better quantify the performance on capturing nuanced colors, we designed a new metric based on color statistics for the task of concept editing as an additional quantitative evaluation.
>
> Specifically, we compare the color of a 16 × 16 patch cropped around the center (focusing on the object) from both the input image and the image generated by the method being evaluated. The original size of the image is  64 × 64. We then compute the MSE of the mean color of the two patches, and report the average of the metric across all examples in Figure 11 in Table 1. Results suggest that our method captures colors more accurately compared to the baselines.
>
>
> **Q3 - Why concept embeddings don’t collapse to word embeddings?**
>
> During training time, the anchor loss (Equation (2)) encourages the encoder predictions to converge to a *meaningful* subspace within the word embedding space [1]. This ensures that these embeddings can be readily visualized by a pre-trained text-to-image generation model, and improves the compositionality across different concept axes, as shown in Figure 8 in the main paper.
>
> The anchor loss is only a *soft* constraint with a small weight in addition to the reconstruction loss (Equation (1)) and therefore does not enforce the concept embedding predictions to be identical to the word embeddings. Empirically this is also the case, as shown in Figure 8. In these figures, we compare the concept embeddings predicted by the `color` encoder to the text embedding of the training-time BLIP-2 label, e.g. “blue” from Figure 8, and the former preserves the specific color of the input image while the latter does not.
>
> [1] Encoder-based Domain Tuning for Fast Personalization of Text-to-Image Models. Rinon Gal, Moab Arar, Yuval Atzmon, Amit H. Bermano, Gal Chechik, Daniel Cohen-Or. 2023.
>
>
> **Q4 - Textual Inversion vs. directly using text inputs**
>
> Thanks for the suggestion. We included the following discussion in the appendix:
> > As discussed in Section 3.1, compared to directly using text as inputs for image generation, using techniques like Textual Inversion (Gal et al., 2022), our model is able to capture more nuanced visual details of a particular image with continuous embeddings, instead of discrete words. This can be illustrated in the empirical results in Figures 11 and 12 as well as the results in the main paper, which show that our method preserves the nuances from input images more accurately than the BLIP-2 baseline which uses texts for conditional generation.
>
>
> **Q5 - Other clarifications.**
>
> - We have updated Table 1 in the main paper to make sure the CLIP-based scores and human evaluation scores are clearly separated.
> - We’ve included the proper citation for DeepFloyd. We apologize for missing it.
> - Thanks for the reference! We’ve included a discussion in the related work section.
> - The quantitative evaluation for the BLIP-based baseline is added to Table 1.

---

### Author Response · Authors · 2023-11-23
**General Response**

We appreciate the valuable feedback from all the reviewers. We are glad that reviewers found our idea interesting (`qJGK`), our framework efficient (`dyyk`) and simple yet effective (`jNDi`), and the paper well-written (`jNDi`) with Good use of diagrams (`qJGK`).

In this response, we provide:
1. additional inference *results on real-world images* (`qJGK`,`dyyk`);
2. new experiments on new concept axes extension *without retraining existing encoders* (`dyyk`);
3. additional quantitative evaluation results (`qJGK`,`jNDi`);
4. more qualitative results for baseline comparisons (`jNDi`);
5. further clarifications.

We include visual results for rebuttal in the `Rebuttal PDF` which is uploaded through the “Supplementary Material” button for easy reference [here](https://openreview.net/attachment?id=juuyW8B8ig&name=supplementary_material). In the responses below, we refer to figures and tables in this `Rebuttal PDF`, unless otherwise specified.

All new materials and changes have also been updated in the full manuscript and appendix marked in blue [here](https://openreview.net/pdf?id=juuyW8B8ig).



**Q1 - Inference Results on Real-world images (`qJGK`,`dyyk`)**

We provide new inference results on real-world images in Figures 1 to 5 in the `Rebuttal PDF`, spanning various types of objects, including furniture, kitchen utensils, fruits, and artwork. Despite *only* being trained on images generated by diffusion-based models, the concept encoders generalize well to diverse, complex real-world images, including *unseen* types of objects, materials, colors, and styles. Note that some of the examples originally presented in the main paper also came from real photos, such as Figure 5 from the main paper.


**Q2 - Extending with new concept axes (`dyyk`)**

Our model can be extended with more concept axes by training new concept encoders *without retraining the existing ones*, as shown in Figures 9 and 10. Given a model trained with concept encoders for `category` and `color`, we additionally train a third encoder for `material` while keeping the other two encoders *frozen*. With this progressive training procedure, the model is able to extend to new concept axes while maintaining the disentanglement of the frozen encoders.


**Q3 - Additional quantitative evaluation (`qJGK`,`jNDi`)**

To better quantify the performance on capturing nuanced colors, we designed a new metric based on color statistics for the task of concept editing as an additional quantitative evaluation.

Specifically, we compare the color of a 16 × 16 patch cropped around the center (focusing on the object) from both the input image and the image generated by the method being evaluated. The original size of the image is  64 × 64. We then compute the MSE of the mean color of the two patches, and report the average of the metric across all examples in Figure 11 in Table 1. Results suggest that our method captures colors more accurately compared to the baselines.


**Q4 - Additional visual results corresponding to the CLIP-Score evaluation (`jNDi`)**

In addition to the results in the main paper, in Figure 11 of the `Rebuttal PDF`, we provide more qualitative examples accompanying the experiments in Table 1. From these visual examples, we observed that the color nuances captured by ours are more accurate compared to the BLIP-2 baseline. However, since the CLIP-based metric specified in Section 4.3 of the paper cannot fully capture the minute differences, the BLIP-2 baseline still achieves comparable scores to our method despite this evident gap in visual results. To better quantify such visual differences in colors, we further designed a new metric based on color statistics, as explained in Q3 above.


**Q5 - Include BLIP-based baseline results also in Table 1 (`qJGK`)**

Thanks for the suggestion. We have included the numbers for the BLIP-based baseline in Table 1. In particular, we reran the human study with 20 users including this baseline in the comparison, and updated the results in the Table (copied below), which show significantly better editing quality. Note that since the human study measures the relative performance of all baselines, the scores for all methods are updated compared to the version in the original paper.

| Method          	|  Edit Category |  Edit Color |
|:--------------------|:--------------:|:-----------:|
| Null-text Inversion | 0.287     	| 0.316 |
| InstructPix2Pix 	| 0.233      	| 0.648	|
| Baseline w/ BLIP-2  | 0.448      	| 0.379	|
| Ours            	| **0.968**     	|**0.840**	|


**Q6 - Interpolation results**

We further demonstrate concept interpolation results. By interpolating between two concept embeddings, our model can generate meaningful images depicting gradual changes from one concept to another, such as meaningful hybrids of cherries and bananas shown in Figures 6 and 7. The details of the interpolation procedure are specified in Section 2 of the `Rebuttal PDF`.

---

### Meta-Review · Area_Chair_HhMQ · 2023-12-10

**Metareview:**

**Summary**

This submission proposes a new framework for disentangle and compose visual concept. The authors define visual concepts as a set of K axes such as category, color, and style, each with values that may be difficult to describe with language. More specifically, visual concepts are distilled from a text-to-image model, using synthetic datasets generated from the text-to-image model itself. A VQA model is leveraged to automatically obtain “concept anchors” to promote disentanglement.

**Strengths**

- Well-written paper
- Simple and effective method. Fast to train with good image generation quality.
- Does not require human annotation.
- Compared to similar work such as textual inversion, the proposed approach does not require re-training for each inference example.
- The idea of using VQA model to produce anchors is novel and interesting


**Weaknesses**

- Using synthetic images. Experiments on real images was provided in rebuttal
- Limited quantitative evaluation. Additional evaluation was done during rebuttal
- Not clear how the learned concept is useful. It seems to be limited to small scale composable image generation in a research setting. Not sure how can this framework scale to real-world application, similar to the modern high-quality text-to-image generation.


**AC’s additional comments**

In my opinion, this paper is borderline acceptable. The idea is novel, and the execution is done flawlessly. However, as an empirical paper in a small-scale, synthetic data focused setting, I’m not sure if it can result in larger impact. My final decision is accept, since two of the reviewers find this paper has some values, and I personally like the idea to disentangle visual concepts from T2I and VQA models. However, I’d remain cautious about practical usage of the proposed method, since the number of concept axes K is very small, and larger K requires training more encoders. I would suggest the authors to discuss potential techniques and practical consideration on how to scale up K, to allow real-world applications.

Two reviewers giving negative reviews did not respond to the rebuttal and reviewer-AC discussion. Their reviews are hence downweighed in my decision process.

Besides, I'd like to suggest the authors to engage in discussions earlier in the future. The responses were posted near the end of discussion period, which left very short time for the reviewers to continue the discussion about unresolved concerns.

**Justification For Why Not Higher Score:**

From the reviews and my own reading of this submission, this paper is novel and interesting. However, I don't have confidence that it can achieve strong impact / contribution. Therefore, I recommend accept as a poster.

**Justification For Why Not Lower Score:**

This paper's core idea is novel, and empirical results are convincing. Instead of following traditional settings and work on incremental techniques, the authors did a great job defining a new problem of disentangled visual concept learning for image generation. Therefore, I recommend accept.

---

### Decision · Program_Chairs · 2024-01-16

Accept (poster)